# *TinyTrain*: Deep Neural Network Training at the Extreme Edge

## Abstract

On-device training is essential for user personalisation and privacy. With the pervasiveness of IoT devices and microcontroller units (MCU), this task becomes more challenging due to the constrained memory and compute resources, and the limited availability of labelled user data. Nonetheless, prior works neglect the data scarcity issue, require excessively long training time (*e.g.* a few hours), or induce substantial accuracy loss ($\geq 10\%$). We propose *TinyTrain*, an on-device training approach that drastically reduces training time by selectively updating parts of the model and explicitly coping with data scarcity. *TinyTrain* introduces a task-adaptive sparse-update method that dynamically selects the layer/channel based on a multi-objective criterion that jointly captures user data, the memory, and the compute capabilities of the target device, leading to high accuracy on unseen tasks with reduced computation and memory footprint. *TinyTrain* outperforms vanilla fine-tuning of the entire network by 3.6-5.0% in accuracy, while reducing the backward-pass memory and computation cost by up to $1,098\times$ and $7.68\times$, respectively. Targeting broadly used real-world edge devices, *TinyTrain* achieves $9.5\times$ faster and $3.5\times$ more energy-efficient training over status-quo approaches, and $2.23\times$ smaller memory footprint than SOTA approaches, while remaining within the 1 MB memory envelope of MCU-grade platforms.

## 1 Introduction

On-device training of deep neural networks (DNNs) on edge devices has the potential to enable diverse real-world applications to *dynamically adapt* to new tasks (Parisi et al., 2019) and different (*i.e.* cross-domain/out-of-domain) data distributions from users (*e.g.* personalisation) (Pan and Yang, 2010), without jeopardising privacy over sensitive data (*e.g.* healthcare) (Gim and Ko, 2022).

Despite its benefits, several challenges hinder the broader adoption of on-device training. **Firstly,** labelled user data are neither abundant nor readily available in real-world IoT applications. **Secondly,** edge devices are often characterised by severely limited memory. With the *forward* and *backward passes* of DNN training being significantly memory-hungry, there is a mismatch between memory requirements and memory availability at the extreme edge. Even architectures tailored to microcontroller units (MCUs), such as MCUNet (Lin et al., 2020), require almost 1 GB of training-time memory (see Table 2), which far exceeds the RAM size of widely used embedded devices, such as Raspberry Pi Zero 2 (512 MB), and commodity MCUs (1 MB). **Lastly,** on-device training is limited by the constrained processing capabilities of edge devices, with training requiring at least $3\times$ more computation (*i.e.* multiply-accumulate (MAC) count) than inference (Xu et al., 2022). This places an excessive burden on tiny edge devices that host less powerful CPUs, compared to the server-grade CPUs or GPUs (Lin et al., 2022).

Recently, on-device training works have been proposed. These, however, have limitations. First, *fine-tuning* only the last layer (Lee and Nirjon, 2020; Ren et al., 2021) leads to considerable accuracy loss (>10%) that far exceeds the typical drop tolerance. Moreover, memory-saving techniques by means of *recomputation* (Chen et al., 2016; Patil et al., 2022; Wang et al., 2022; Gim and Ko, 2022) that trade-off more computation for lower memory usage, incur significant computation overhead, further increasing the already excessive on-device training time. Lastly, *sparse-update* methods (Profentzas et al., 2022; Lin et al., 2022; Cai et al., 2020) selectively update only a subset of layers and channels during on-device training, effectively reducing both memory and computation

loads. Nonetheless, as shown in §3.2, the performance of these approaches drops dramatically (up to 7.7% for *SparseUpdate* (Lin et al., 2022)) when applied at the extreme edge where data availability is low. Also, these methods require running *a few thousands of* computationally heavy search (Lin et al., 2022) or pruning (Profentzas et al., 2022) processes on powerful GPUs to identify important layers/channels for each target dataset, unable to adapt to the properties of the user data on the fly.

To address the aforementioned challenges and limitations, we present *TinyTrain*, the first approach that fully enables compute-, memory-, and data-efficient on-device training on constrained edge devices. *TinyTrain* departs from the static configuration of the sparse-update policy, *i.e.* the subset of layers and channels to be fine-tuned being fixed, and proposes *task-adaptive sparse update*. Our task-adaptive sparse update requires running *only once* for each target dataset and can be efficiently executed on resource-constrained edge devices. This enables us to adapt the layer/channel selection in a task-adaptive manner, leading to better on-device adaptation and higher accuracy. Specifically, we introduce a novel *multi-objective criterion* to guide the layer/channel selection process that captures both the importance of channels and their computational and memory cost. Then, at run time, we propose a *dynamic layer/channel selection* scheme that dynamically adapts the sparse update policy using our multi-objective criterion. As *TinyTrain* takes into account both the properties of the user data, and the memory and processing capacity of the target device, *TinyTrain* enables on-device training with a significant reduction in memory and computation without accuracy loss over the state-of-the-art (SOTA) (Lin et al., 2022). Finally, to further address the

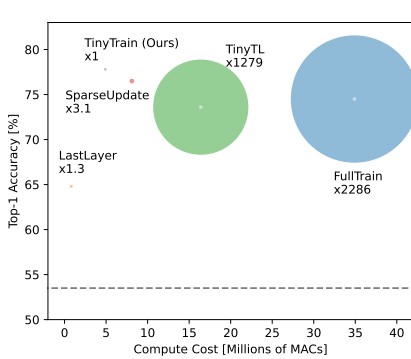

Figure 1: Cross-domain accuracy (y-axis) and compute cost in MAC count (x-axis) of *TinyTrain* and existing methods, targeting MobileNetV2 on Meta-Dataset. The radius of the circles and the corresponding text denote the increase in the memory footprint of each baseline over *TinyTrain*. The dotted line represents the accuracy without on-device training.

drawbacks of data scarcity, *TinyTrain* enhances the conventional on-device training pipeline by means of a few-shot learning (FSL) pre-training scheme; this step meta-learns a reasonable global representation that allows on-device training to be sample-efficient and reach high accuracy despite the limited and cross-domain target data.

Figure 1 presents a comparison of our method's performance with existing on-device training approaches. *TinyTrain* achieves the highest accuracy, with gains of 3.6-5.0% over fine-tuning the entire DNN, denoted by *FullTrain*. On the compute front, *TinyTrain* significantly reduces the memory footprint and computation required for backward pass by up to 1,098× and 7.68×, respectively. *TinyTrain* further outperforms the SOTA *SparseUpdate* method in all aspects, yielding: (a) 2.6-7.7% accuracy gain across nine datasets; (b) 1.59-2.23× reduction in memory; and (c) 1.52-1.82× lower computation costs. Finally, we demonstrate how our work makes important steps towards efficient training on very constrained edge devices by deploying *TinyTrain* on Raspberry Pi Zero 2 and Jetson Nano and showing that our multi-objective criterion can be efficiently computed within 20-35 seconds on both of our target edge devices (*i.e.* 3.4-3.8% of the total training time of *TinyTrain*), removing the necessity of offline search process of important layers and channels. Also, *TinyTrain* achieves an end-to-end on-device training in 10 minutes, an order of magnitude speedup over the two-hour training of *FullTrain* on Pi Zero 2. These findings open the door, for the first time, to performing on-device training with acceptable performance on a variety of resource-constrained devices, such as MCUs embedded in IoT frameworks.

## 2 METHODOLOGY

**Problem Formulation.** From a learning perspective, on-device DNN training at the extreme edge imposes unique characteristics that the model needs to address during deployment, primarily: (1) unseen target tasks with different data distributions (cross-domain), and (2) scarce labelled user data. To formally capture this setting, in this work, we cast it as a cross-domain few-shot learning (CDFSL) problem. In particular, we formulate it as *K-way-N-shot* learning (Triantafillou et al., 2020) which allows us to accommodate more general scenarios instead of optimising towards one specific CDFSL setup (*e.g.* 5-way 5-shots). This formulation requires us to learn a DNN for $K$ classes given $N$ sam-

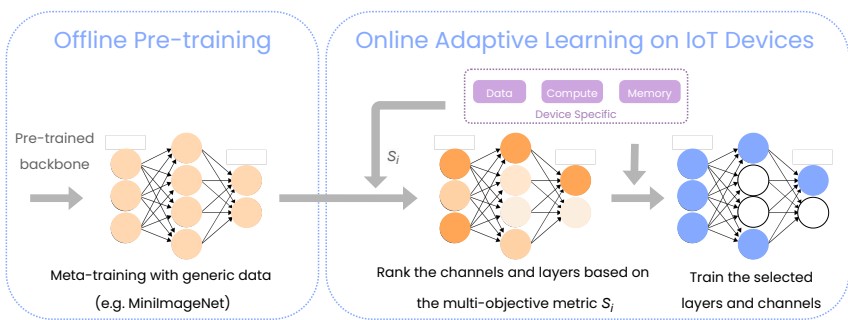

Figure 2: The overview of *TinyTrain*. It consists of (1) offline pre-training and (2) online adaptive learning stages. In (1), *TinyTrain* pre-trains and meta-trains DNNs to improve the attainable accuracy when only a few data are available for adaptation. Then, in (2), *TinyTrain* performs task-adaptive sparse-update based on the multi-objective criterion and dynamic layer/channel selection that co-optimises both memory and computations.

ples per class. To further push towards realistic scenarios, we learn *one* global DNN representation from various $K$ and $N$, which can be used to learn novel tasks (see §3.1 and §A.1 for details).

**Our Pipeline.** Figure 2 shows the processing flow of *TinyTrain* comprising two stages. The first stage is *offline learning*. By means of pre-training and meta-training, *TinyTrain* aims to find an informed weight initialisation, such that subsequently the model can be rapidly adapted to the user data with only a few samples (5-30), drastically reducing the burden of manual labelling and the overall training time compared to state of the art methods. The second stage is *online learning*. This stage takes place on the target edge device, where *TinyTrain* utilises its task-adaptive sparse-update method to selectively fine-tune the model using the limited user-specific, cross-domain target data, while minimising the memory and compute overhead.

## 2.1 Few-Shot Learning-Based Pre-training

The vast majority of existing on-device training pipelines optimise certain aspects of the system (*i.e.* memory or compute) via memory-saving techniques (Chen et al., 2016; Patil et al., 2022; Wang et al., 2022; Gim and Ko, 2022) or fine-tuning a small set of layers/channels (Cai et al., 2020; Lin et al., 2022; Ren et al., 2021; Lee and Nirjon, 2020; Profentzas et al., 2022). However, these methods neglect the aspect of sample efficiency in the low-data regime of tiny edge devices. As the availability of labelled data is severely limited at the extreme edge, existing on-device training approaches suffer from insufficient learning capabilities under such conditions.

In our work, we depart from the transfer-learning paradigm (*i.e.* DNN pre-training on source data, followed by fine-tuning on target data) of existing on-device training methods that are unsuitable to the very low data regime of edge devices. Building upon the insight of recent studies (Hu et al., 2022) that transfer learning does not reach a model's maximum capacity on unseen tasks in the presence of only limited labelled data, we augment the *offline* stage of our training pipeline as follows. Starting from the *pre-training* of the DNN backbone using a large-scale public dataset, we introduce a subsequent *meta-training* process that meta-trains the pre-trained DNN given only a few samples (5-30) per class on simulated tasks in an episodic fashion. As shown in §3.3, this approach enables the resulting DNNs to perform more robustly and achieve higher accuracy when adapted to a target task despite the low number of examples, matching the needs of tiny edge devices. As a result, our few-shot learning (FSL)-based pre-training constitutes an important component to improve the accuracy given only a few samples for adaptation, reducing the training time while improving data and computation efficiency. Thus, *TinyTrain* alleviates the drawbacks of current work, by explicitly addressing the lack of labelled user data, and achieving faster training and lower accuracy loss.

**Pre-training.** For the backbones of our models, we employ feature extractors of different DNN architectures as in §3.1. These feature backbones are pre-trained with a large-scale image dataset, *e.g.* ImageNet (Deng et al., 2009).

**Meta-training.** For the meta-training phase, we employ the metric-based ProtoNet (Snell et al., 2017), which has been demonstrated to be simple and effective as an FSL method. ProtoNet computes

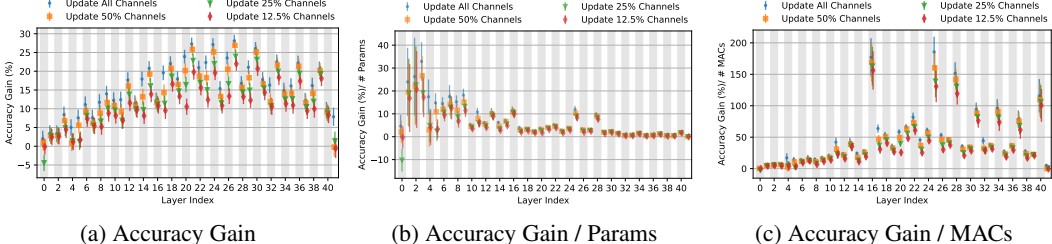

(a) Accuracy Gain      (b) Accuracy Gain / Params      (c) Accuracy Gain / MACs

Figure 3: Memory- and compute-aware analysis of MCUNet by updating four different channel ratios on each layer. (a) Accuracy gain per layer is generally highest on the first layer of each block. (b) Accuracy gain per parameter of each layer is higher on the second layer of each block, but it is not a clear pattern. (c) Accuracy gain per MACs of each layer has peaked on the second layer of each block. These observations show accuracy, memory footprint, and computes in a trade-off relation.

the class centroids (*i.e.* prototypes) for a given support set and then performs nearest-centroid classification using the query set. Specifically, given a pre-trained feature backbone $f$ that maps inputs $x$ to an $m$-dimensional feature space, ProtoNet first computes the prototypes $c_k$ for each class $k$ on the support set as $c_k = \frac{1}{N_k} \sum_{i:y_i=k} f(x_i)$, where $N_k = \sum_{i:y_i=k} 1$ and $y$ are the labels. The probability of query set inputs $x$ for each class $k$ is then computed as:

$$p(y = k|x) = \frac{exp(-d(f(x), c_k))}{\sum_j exp(-d(f(x), c_j))} \tag{1}$$

We use cosine distance as the distance measure $d$ similarly to Hu et al. (2022). Note that ProtoNet enables the *various-way-various-shot* setting since the prototypes can be computed regardless of the number of ways and shots. The feature backbones are meta-trained with MiniImageNet (Vinyals et al., 2016), a commonly used source dataset in CSFSL, to provide a weight initialisation generalisable to multiple downstream tasks in the subsequent online stage (see §F.2 for meta-training cost analysis).

## 2.2 TASK-ADAPTIVE SPARSE UPDATE

Existing FSL pipelines generally focus on data and sample efficiency and attend less to system optimisation (Finn et al., 2017; Snell et al., 2017; Hospedales et al., 2022; Triantafillou et al., 2020; Hu et al., 2022), rendering most of these algorithms undeployable for the extreme edge, due to high computational and memory costs. In this context, sparse update (Lin et al., 2022; Profentzas et al., 2022), which dictates that only a subset of essential layers and channels are to be trained, has emerged as a promising paradigm for making training feasible on resource-constrained devices.

Two key design decisions of sparse-update methods are *i)* the scheme for determining the *sparse-update policy*, *i.e.* which layers/channels should be fine-tuned, and *ii)* how often should the sparse-update policy be modified. In this context, a SOTA method, such as *SparseUpdate* (Lin et al., 2022), is characterised by important limitations. First, it casts the layer/channel selection as an optimisation problem that aims to maximise the accuracy gain subject to the memory constraints of the target device. However, as the optimisation problem is combinatorial, *SparseUpdate* solves it offline by means of a heuristic evolutionary algorithm that requires *a few thousand trials*. Second, as the search process for a good sparse-update policy is too costly, it is practically infeasible to dynamically adjust the sparse-update policy whenever new target datasets are given, leading to performance degradation.

**Multi-Objective Criterion.** With resource constraints being at the forefront in tiny edge devices, we investigate the trade-offs among accuracy gain, compute and memory cost. To this end, we analyse each layer's contribution (*i.e.* accuracy gain) on the target dataset by updating a single layer at a time, together with cost-normalised metrics, including accuracy gain *per parameter* and *per MAC operation* of each layer. Figure 3 shows the results of MCUNet (Lin et al., 2020) on the Traffic Sign (Houben et al., 2013) dataset (see §E.1 for more results). We make the following observations: (1) the peak point of accuracy gain occurs at the first layer of each block (pointwise convolutional layer) (Figure 3a), (2) the accuracy gain per parameter and computation cost occurs at the second layer of each block (depthwise convolutional layer) (Figures 3b and 3c). These findings indicate a

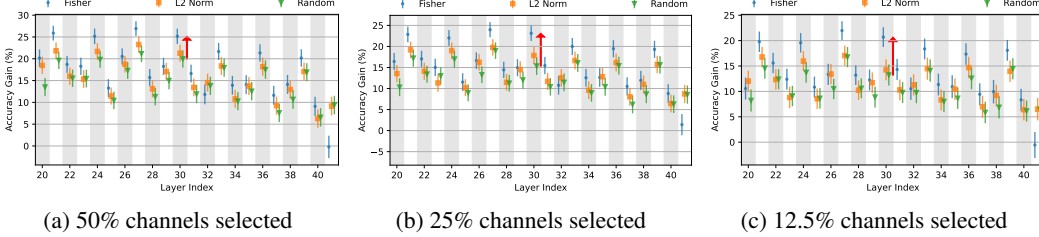

(a) 50% channels selected      (b) 25% channels selected      (c) 12.5% channels selected

Figure 4: The pairwise comparison between our dynamic channel selection and static channel selections (*i.e.* Random and L2-Norm) on MCUNet. The dynamic channel selection consistently outperforms static channel selections as the accuracy gain per layer differs by up to 8%. Also, the gap between dynamic and static channel selections increases as fewer channels are selected for updates.

non-trivial trade-off between accuracy, memory, and computation, demonstrating the necessity for an effective layer/channel selection method that jointly considers all the aspects.

To encompass both accuracy and efficiency aspects, we design a multi-objective criterion for the layer selection process of our task-adaptive sparse-update method. To quantify the importance of channels and layers on the fly, we propose the use of Fisher information on activations (Amari, 1998; Theis et al., 2018; Kim et al., 2022), often used to identify *less important* channels/layers for pruning (Theis et al., 2018; Turner et al., 2020). Whereas we use it as a proxy for *more important* channels/layers for weight update. Formally, given $N$ examples of target inputs, the Fisher information $\Delta_o$ can be calculated after backpropagating the loss $L$ with respect to activations $a$ of a layer:

$$\Delta_o = \frac{1}{2N} \sum_n^N (\sum_d^D a_{nd} g_{nd})^2 \qquad (2)$$

where gradient is denoted by $g_{nd}$ and $D$ is feature dimension of each channel (*e.g.* $D = H \times W$ of height $H$ and width $W$). We obtain the Fisher potential $P$ for a whole layer by summing $\Delta_o$ for all activation channels as: $P = \sum_o \Delta_o$.

Having established the importance of channels in each layer, we define a new multi-objective metric $s$ that jointly captures importance, memory footprint and computational cost:

$$s_i = \frac{P_i}{\frac{\|W_i\|}{\max\limits_{l \in \mathcal{L}}(\|W_l\|)} \times \frac{M_i}{\max\limits_{l \in \mathcal{L}}(M_l)}} \qquad (3)$$

where $\|W_i\|$ and $M_i$ represent the number of parameters and multiply-accumulate (MAC) operations of the $i$-th layer and are normalised by the respective maximum values $\max\limits_{l \in \mathcal{L}}(\|W_l\|)$ and $\max\limits_{l \in \mathcal{L}}(M_l)$ across all layers $\mathcal{L}$ of the model. This multi-objective metric enables *TinyTrain* to rank different layers and prioritise the ones with higher Fisher potential per parameter and per MAC during layer selection.

**Dynamic Layer/Channel Selection.** We now present our *dynamic layer/channel selection* scheme, the second component of our task-adaptive sparse update, that runs at the *online* learning stage (*i.e.* deployment and meta-testing phase). Concretely, with reference to Algorithm 1, when a new on-device task needs to be learned (*e.g.* a new user or dataset), the sparse-update policy is modified to match its properties (lines 1-4). Contrary to the existing layer/channel selection approaches that remain fixed across tasks, our method is based on the key insight that different features/channels can play a more important role depending on the target dataset/task/user. As shown in §3.3, effectively tailoring the layer/channel selection to each task leads to superior accuracy compared to the pre-determined, static layer selection scheme of *SparseUpdate*, while further minimising system overheads.

As an initialisation step, *TinyTrain* is first provided with the memory and computation budget determined by hardware and users, *e.g.* around 1 MB and 15% of total MACs can be given as backward-pass memory and computational budget. Then, we calculate the Fisher potential for each convolutional layer using the given inputs of a target task efficiently (refer to §F.1 for further details) (lines 1-2). Then, based on our multi-objective criterion (Eq. (3)) (line 3), we score each layer and progressively select as many layers as possible without violating the memory constraints (imposed by

---

**Algorithm 1:** Online learning stage of *TinyTrain*

---

**Require:** Meta-trained backbone weights $W$, iterations $k$, Train data $D_{\text{train}}$, Test data $D_{\text{test}}$,
memory and compute budgets $B_{\text{mem}}, B_{\text{compute}}$

/* Dynamic layer/channel selection                                              */

1  Compute the gradient using the given samples $D_{\text{train}}$
2  Compute the Fisher potential using Eq. (2) from the Fisher information
3  Compute our multi-objective metric $s$ using Eq. (3)
4  Perform the dynamic layer & channel selection using $\{W, s, B_{\text{mem}}, B_{\text{compute}}\}$

/* Perform sparse fine-tuning                                                    */

5  **for** $t = 1, ..., k$ **do**
6  $\quad$ Update the selected layers/channels using $D_{\text{train}}$

7  Evaluate the fine-tuned backbone using $D_{\text{test}}$

---

the memory usage of the model, optimiser, and activations memory) and resource budgets (imposed by users and target hardware) on an edge device (line 4).

After having selected layers, within each selected layer, we identify the top-$K$ most important channels to update using the Fisher information for each activation channel, $\Delta_o$, that was precomputed during the initialisation step (line 4). Note that the overhead of our dynamic layer/channel selection is minimal, which takes only 20-35 seconds on edge devices (more analysis in §3.2 and §3.3). Having finalised the layer/channel selection, we proceed with their sparse fine-tuning of the meta-trained DNN during meta-testing (see §C for detailed procedures). As in Figure 4 (MCUNet on Traffic Sign; refer to §E.5 for more results), dynamically identifying important channels for an update for each target task outperforms the static channel selections such as random- and L2-Norm-based selection.

Further, as *TinyTrain* requires to run the dynamic layer/channel selection *only once* for each target dataset by obtaining multi-objective criterion, *TinyTrain* effectively alleviates the burdens of running the computationally heavy search processes *a few thousand times*.

*Overall, the dynamic layer/channel selection scheme facilitates TinyTrain to achieve superior accuracy, while further minimising the memory and computation cost by co-optimising both system constraints, thereby enabling memory- and compute-efficient training at the extreme edge.*

## 3 EVALUATION

### 3.1 EXPERIMENTAL SETUP

We briefly explain our experimental setup in this subsection (refer to §A for further details).

**Datasets.** We use *MiniImageNet* (Vinyals et al., 2016) as the *meta-train dataset*, following the same setting as prior works on cross-domain FSL (Hu et al., 2022; Triantafillou et al., 2020). For *meta-test datasets* (*i.e.* target datasets of different domains than the source dataset of MiniImageNet), we employ all nine out-of-domain datasets of various domains from Meta-Dataset (Triantafillou et al., 2020), excluding ImageNet because it is used to pre-train the models before deployment, making it an in-domain dataset. Extensive experimental results with nine different *cross-domain* datasets showcase the robustness and generality of our approach to the challenging CDFSL problem.

**Architectures.** Following Lin et al. (2022), we employ three DNN architectures: *MCUNet* (Lin et al., 2020), *MobileNetV2* (Sandler et al., 2018), and *ProxylessNAS* (Cai et al., 2019). The models are pretrained with ImageNet and optimised for resource-limited IoT devices by adjusting width multipliers.

**Evaluation.** To evaluate the CDFSL performance, we sample 200 tasks from the test split for each dataset. Then, we use testing accuracy on unseen samples of a new-domain target dataset. Following Triantafillou et al. (2020), the number of classes and support/query sets are sampled uniformly at random regarding the dataset specifications. On the computational front, we present the computation cost in MAC operations and the memory usage. We measure latency and energy consumption when running end-to-end DNN training on actual edge devices (see §D for system implementation).

**Baselines.** We compare *TinyTrain* with the following five baselines: (1) *None* does not perform any on-device training; (2) *FullTrain* (Pan and Yang, 2010) fine-tunes the entire model, representing a conventional transfer-learning approach; (3) *LastLayer* (Ren et al., 2021; Lee and Nirjon, 2020)

Table 1: Top-1 accuracy results of *TinyTrain* and the baselines. *TinyTrain* achieves the highest accuracy with three DNN architectures on nine cross-domain datasets.

| Model | Method | Traffic | Omniglot | Aircraft | Flower | CUB | DTD | QDraw | Fungi | COCO | Avg. |
|---|---|---|---|---|---|---|---|---|---|---|---|
| MCUNet | None | 35.5 | 42.3 | 42.1 | 73.8 | 48.4 | 60.1 | 40.9 | 30.9 | 26.8 | 44.5 |
| | FullTrain | **82.0** | 72.7 | 75.3 | 90.7 | 66.4 | 74.6 | 64.0 | 40.4 | 36.0 | 66.9 |
| | LastLayer | 55.3 | 47.5 | 56.7 | 83.9 | 54.0 | 72.0 | 50.3 | 36.4 | 35.2 | 54.6 |
| | TinyTL | 78.9 | 73.6 | 74.4 | 88.6 | 60.9 | 73.3 | 67.2 | 41.1 | 36.9 | 66.1 |
| | SparseUpdate | 72.8 | 67.4 | 69.0 | 88.3 | 67.1 | 73.2 | 61.9 | 41.5 | 37.5 | 64.3 |
| | *TinyTrain* (Ours) | 79.3 | **73.8** | **78.8** | **93.3** | **69.9** | **76.0** | 67.3 | **45.5** | **39.4** | **69.3** |
| Mobile NetV2 | None | 39.9 | 44.4 | 48.4 | 81.5 | 61.1 | 70.3 | 45.5 | 38.6 | 35.8 | 51.7 |
| | FullTrain | 75.5 | 69.1 | 68.9 | 84.4 | 61.8 | 71.3 | 60.6 | 37.7 | 35.1 | 62.7 |
| | LastLayer | 58.2 | 55.1 | 59.6 | 86.3 | 61.8 | 72.2 | 53.3 | 39.8 | 36.7 | 58.1 |
| | TinyTL | 71.3 | 69.0 | 68.1 | 85.9 | 57.2 | 70.9 | **62.5** | 38.2 | 36.3 | 62.1 |
| | SparseUpdate | 77.3 | **69.1** | 72.4 | 87.3 | 62.5 | 71.1 | 61.8 | 38.8 | 35.8 | 64.0 |
| | *TinyTrain* (Ours) | **77.4** | 68.1 | **74.1** | **91.6** | **64.3** | **74.9** | 60.6 | **40.8** | **39.1** | **65.6** |
| Proxyless NASNet | None | 42.6 | 50.5 | 41.4 | 80.5 | 53.2 | 69.1 | 47.3 | 36.4 | 38.6 | 51.1 |
| | FullTrain | 78.4 | 73.3 | 71.4 | 86.3 | 64.5 | 71.7 | 63.8 | 38.9 | 37.2 | 65.0 |
| | LastLayer | 57.1 | 58.8 | 52.7 | 85.5 | 56.1 | 72.9 | 53.0 | 38.6 | 38.7 | 57.0 |
| | TinyTL | 72.5 | **73.6** | 70.3 | 86.2 | 57.4 | 71.0 | 65.8 | 38.6 | 37.6 | 63.7 |
| | SparseUpdate | 76.0 | 72.4 | 71.2 | 87.8 | 62.1 | 71.7 | 64.1 | 39.6 | 37.1 | 64.7 |
| | *TinyTrain* (Ours) | **79.0** | 71.9 | **76.7** | **92.7** | **67.4** | **76.0** | 65.9 | **43.4** | **41.6** | **68.3** |

updates the last layer only; (4) *TinyTL* (Cai et al., 2020) updates the augmented lite-residual modules while freezing the backbone; and (5) *SparseUpdate* of MCUNetV3 (Lin et al., 2022), is a prior state-of-the-art (SOTA) method for on-device training that statically pre-determines which layers and channels to update before deployment and then updates them online.

## 3.2 MAIN RESULTS

**Accuracy.** Table 1 summarises accuracy results of *TinyTrain* and various baselines after adapting to cross-domain target datasets, averaged over 200 runs. *None* attains the lowest accuracy among all the baselines, demonstrating the importance of on-device training when domain shift in train-test data distribution is present. *LastLayer* improves upon *None* with a marginal accuracy increase, suggesting that updating the last layer is insufficient to achieve high accuracy in cross-domain scenarios, likely due to final layer limits in the capacity. In addition, *Full-Train*, serving as a strong baseline as it assumes unlimited system resources, achieves high accuracy. *TinyTL* also yields moderate accuracy. However, as both *FullTrain* and *TinyTL* require prohibitively large memory and computation for training, they remain unsuitable to operate on resource-constrained devices, as shown below.

Table 2: Comparison of the memory footprint and computation cost for a backward pass.

| Model | Method | Memory | Ratio | Compute | Ratio |
|---|---|---|---|---|---|
| MCUNet | FullTrain | 906 MB | 1,013× | 44.9M | 6.89× |
| | LastLayer | 2.03 MB | 2.27× | 1.57M | 0.23× |
| | TinyTL | 542 MB | 606× | 26.4M | 4.05× |
| | SparseUpdate | 1.43 MB | 1.59× | 11.9M | 1.82× |
| | *TinyTrain* (Ours) | **0.89 MB** | 1× | **6.51M** | 1× |
| Mobile NetV2 | FullTrain | 1,049 MB | 987× | 34.9M | 7.12× |
| | LastLayer | 1.64 MB | 1.54× | 0.80M | 0.16× |
| | TinyTL | 587 MB | 552× | 16.4M | 3.35× |
| | SparseUpdate | 2.08 MB | 1.96× | 8.10M | 1.65× |
| | *TinyTrain* (Ours) | **1.06 MB** | 1× | **4.90M** | 1× |
| Proxyless NASNet | FullTrain | 857 MB | 1,098× | 38.4M | 7.68× |
| | LastLayer | 1.06 MB | 1.36× | 0.59M | 0.12× |
| | TinyTL | 541 MB | 692× | 17.8M | 3.57× |
| | SparseUpdate | 1.74 MB | 2.23× | 7.60M | 1.52× |
| | *TinyTrain* (Ours) | **0.78 MB** | 1× | **5.00M** | 1× |

*TinyTrain* achieves the best accuracy on most datasets and the highest average accuracy across them, outperforming all the baselines including *FullTrain*, *LastLayer*, *TinyTL*, and *SparseUpdate* by 3.6-5.0 percentage points (pp), 13.0-26.9 pp, 4.8-7.2 pp, and 2.6-7.7 pp, respectively. This result indicates that our approach of identifying important parameters on the fly in a task-adaptive manner and updating them could be more effective in preventing overfitting given the few samples of CDFSL.

**Memory & Compute.** We investigate the memory and computation costs to perform a backward pass, which takes up the majority of the memory and computation of training (Sohoni et al., 2019; Xu et al., 2022). As shown in Table 2, we first observe that *FullTrain* and *TinyTL* consume significant amounts of memory, ranging between 857-1,049 MB and 541-587 MB, respectively, *i.e.* up to 1,098× and 692× more than *TinyTrain*, which exceeds the typical RAM size of IoT devices, such as Pi Zero (*e.g.* 512 MB). Note that a batch size of 100 is used for these two baselines as their accuracy degrades catastrophically with smaller batch sizes. Conversely, the other methods, including *LastLayer*, *SparseUpdate*, and *TinyTrain*, use a batch size of 1 and yield a smaller memory footprint and

computational cost. Importantly, compared to *SparseUpdate*, *TinyTrain* enables on-device training with 1.59-2.23× less memory and 1.52-1.82× less computation (see §A.4 for details on acquiring memory and compute). This gain can be attributed to the multi-objective criterion of *TinyTrain*'s sparse-update method, which co-optimises both memory and computation. Also, note that evaluating our multi-criterion objective does not incur excessive memory overhead, as detailed in §F.1.

**End-to-End Latency and Energy Consumption.** We now examine the run-time system efficiency by measuring *TinyTrain*'s end-to-end training time and energy consumption. To this end, we deploy *TinyTrain* and the baselines on constrained edge devices, Pi Zero 2 (Figure 5) and Jetson Nano (§E.3). To measure the overall on-device training cost (excluding offline pre-training and meta-training), we include the time and energy consumption: (1) to load a pre-trained model, and (2) to perform training using all the samples (*e.g.* 25) for a certain number of iterations (*e.g.* 40), and (3) to

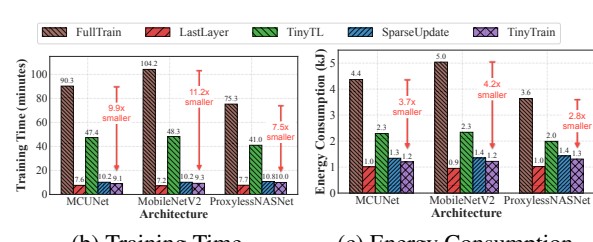

(b) Training Time    (c) Energy Consumption

Figure 5: End-to-end latency and energy consumption of the on-device training methods on three architectures.

perform dynamic layer/channel selection for task-adaptive sparse update (only for *TinyTrain*).

*TinyTrain* yields 1.08-1.12× and 1.3-1.7× faster on-device training than SOTA on Pi Zero 2 and Jetson Nano, respectively. Also, *TinyTrain* completes an end-to-end on-device training process within 10 minutes, an order of magnitude speedup over the two-hour training of conventional transfer learning, a.k.a. *FullTrain* on Pi Zero 2. Moreover, the latency of *TinyTrain* is shorter than all the baselines except for that of *LastLayer* which only updates the last layer but suffers from high accuracy loss. In addition, *TinyTrain* shows a significant reduction in the energy consumption (incurring 1.20-1.31kJ) compared to all the baselines, except for *LastLayer*, similarly to the latency results.

**Summary.** *Our results demonstrate that TinyTrain can effectively learn cross-domain tasks requiring only a few samples, i.e. it generalises well to new samples and classes unseen during the offline learning phase. Furthermore, TinyTrain enables fast and energy-efficient on-device training on constrained IoT devices with significantly reduced memory footprint and computational load.*

## 3.3 ABLATION STUDY AND ANALYSIS

**Impact of Meta-Training.** We compare the accuracy between pre-trained DNNs with and without meta-training using MCUNet. Figure 6a shows that meta-training improves the accuracy by 0.6-31.8 pp over the DNNs without meta-training across all the methods (see §E.4 for more results). For *TinyTrain*, offline meta-training increases accuracy by 5.6 pp on average with reasonable cost (see §F.2 for cost analysis of meta-training). *This result shows the impact of meta-training compared to conventional transfer learning, demonstrating the effectiveness of our FSL-based pre-training (§2.1).*

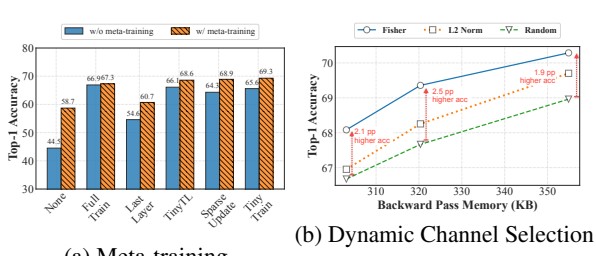

(a) Meta-training

(b) Dynamic Channel Selection

Figure 6: The effect of (a) meta-training and (b) dynamic channel selection using MCUNet averaged over nine cross-domain datasets.

**Robustness of Dynamic Channel Selection.** We compare the accuracy of *TinyTrain* with and without dynamic channel selection, with the same set of layers to be updated within strict memory constraints using MCUNet. This comparison shows how much improvement is derived from dynamically selecting important channels based on our method at deployment time. Figure 6b shows that dynamic channel selection increases accuracy by 0.8-1.7 pp and 1.9-2.5 pp on average compared to static channel selection based on L2-Norm and Random, respectively (see §E.5 for more results). In addition, given a more limited memory budget, our dynamic channel selection maintains higher accuracy than static channel selection. *Our ablation study reveals the robustness of the dynamic channel selection of our task-adaptive sparse-update (§2.2).*

**Efficiency of Task-Adaptive Sparse Update.** Our dynamic layer/channel selection process takes around 20-35 seconds on our employed edge devices (*i.e.* Pi Zero 2 and Jetson Nano), accounting for only 3.4-3.8% of the total training time of *TinyTrain*. Note that our selection process is $30\times$ faster than *SparseUpdate*'s server-based evolutionary search, taking 10 minutes with abundant offline compute resources. *This demonstrates the efficiency of our task-adaptive sparse update.*

## 4 RELATED WORK

**On-Device Training.** Driven by the increasing privacy concerns and the need for post-deployment adaptability to new tasks/users, the research community has recently turned its attention to enabling DNN *training* (*i.e.,* backpropagation having forward and backward passes, and weights update) at the edge. First, researchers proposed memory-saving techniques to resolve the memory constraints of training (Sohoni et al., 2019; Chen et al., 2021; Pan et al., 2021; Evans and Aamodt, 2021; Liu et al., 2022). For example, gradient checkpointing (Chen et al., 2016; Jain et al., 2020; Kirisame et al., 2021) discards activations of some layers in the forward pass and recomputes those activations in the backward pass. Microbatching (Huang et al., 2019) splits a minibatch into smaller subsets that are processed iteratively, to reduce the peak memory needs. Swapping (Huang et al., 2020; Wang et al., 2018; Wolf et al., 2020) offloads activations or weights to an external memory/storage (*e.g.* from GPU to CPU or from an MCU to an SD card). Some works (Patil et al., 2022; Wang et al., 2022; Gim and Ko, 2022) proposed a hybrid approach by combining two or three memory-saving techniques. Although these methods reduce the memory footprint, they incur additional computation overhead on top of the already prohibitively expensive on-device training time at the edge. Instead, our work drastically minimises not only memory but also the amount of computation through its dynamic sparse update that identifies and trains on-the-fly only the most important layers/channels.

A few existing works (Lin et al., 2022; Cai et al., 2020; Profentzas et al., 2022; Qu et al., 2022) have also attempted to optimise both memory and computations, with prominent examples being TinyTL (Cai et al., 2020) and SparseUpdate (Lin et al., 2022). However, TinyTL still demands excessive memory and computation (see §3.2). SparseUpdate suffers from accuracy loss, with a drop of 2.6-7.7% compared to *TinyTrain*) when on-device data are scarce, as at the extreme edge. In contrast, *TinyTrain* enables data-, compute-, and memory-efficient training on tiny edge devices by adopting FSL pre-training and dynamic layer/channel selection.

**Cross-Domain Few-Shot Learning.** Due to the scarcity of labelled user data on the device, developing Few-Shot Learning (FSL) techniques (Hospedales et al., 2022; Finn et al., 2017; Li et al., 2017; Snell et al., 2017; Sung et al., 2018; Satorras and Estrach, 2018; Zhang et al., 2021) is a natural fit for on-device training. Also, a growing body of work focuses on cross-domain (out-of-domain) FSL (CDFSL) (Guo et al., 2020; Hu et al., 2022; Triantafillou et al., 2020) where the source (meta-train) dataset drastically differs from the target (meta-test) dataset. CDFSL is practically relevant since in real-world deployment scenarios, the scarcely annotated target data (*e.g.* earth observation images (Guo et al., 2020; Triantafillou et al., 2020)) is often significantly different from the offline source data (*e.g.* (Mini-)ImageNet). However, FSL-based methods only consider data efficiency, neglecting the memory and computation bottlenecks of on-device training. We explore joint optimisation of all the major bottlenecks of on-device training: data, memory, and computation.

## 5 CONCLUSION

We have developed the first realistic on-device training framework, *TinyTrain*, solving practical challenges in terms of data, memory, and compute constraints for extreme edge devices. *TinyTrain* meta-learns in a few-shot fashion during the offline learning stage and dynamically selects important layers and channels to update during deployment. As a result, *TinyTrain* outperforms all existing on-device training approaches by a large margin enabling, for the first time, fully on-device training on unseen tasks at the extreme edge. It allows applications to generalise to cross-domain tasks using only a few samples and adapt to the dynamics of the user devices and context.

**Limitations & Societal Impacts.** Our evaluation is currently limited to CNN-based architectures on vision tasks. As future work, we hope to extend *TinyTrain* to different architectures (*e.g.* Transformers, RNNs) and applications (*e.g.* audio, biological data). In addition, while on-device training avoids the excessive electricity consumption and carbon emissions of centralised training (Schwartz et al., 2020; Patterson et al., 2022), it has thus far been a significantly draining process for the battery life of edge devices. However, *TinyTrain* paves the way towards alleviating this issue, demonstrated in Figure 5c.

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

# Supplementary Material
## *TinyTrain*: Deep Neural Network Training at the Extreme Edge

# Table of Contents

# A  DETAILED EXPERIMENTAL SETUP

This section provides additional information on the experimental setup.

## A.1  DATASETS

Following the conventional setup for evaluating cross-domain FSL performances on MetaDataset in prior arts (Hu et al., 2022; Triantafillou et al., 2020; Guo et al., 2020), we use *MiniImageNet* (Vinyals et al., 2016) for **Meta-Train** and the non-ILSVRC datasets in MetaDataset (Triantafillou et al., 2020) for **Meta-Test**. Specifically, MiniImageNet contains 100 classes from ImageNet-1k, split into 64 training, 16 validation, and 20 testing classes. The resolution of the images is downsampled to $84 \times 84$. The MetaDataset used as **Meta-Test datasets** consists of nine public image datasets from a variety of domains, namely *Traffic Sign* (Houben et al., 2013), *Omniglot* (Lake et al., 2015), *Aircraft* (Maji et al., 2013), *Flowers* (Nilsback and Zisserman, 2008), CUB (Welinder et al., 2011), DTD (Cimpoi et al., 2014), QDraw (Jongejan et al., 2016), Fungi (Schroeder and Cui, 2018), and COCO (Lin et al., 2014). Note that the ImageNet dataset is excluded as it is already used for pre-training the models during the meta-training phase, which makes it an in-domain dataset. We showcase the robustness and generality of our approach to the challenging cross-domain few-shot learning (CDFSL) problem via extensive evaluation of these datasets. The details of each target dataset employed in our study are described below.

The **Traffic Sign** (Houben et al., 2013) dataset consists of 50,000 images out of 43 classes regarding German road signs.

The **Omniglot** (Lake et al., 2015) dataset has 1,623 handwritten characters (*i.e.* classes) from 50 different alphabets. Each class contains 20 examples.

The **Aircraft** (Maji et al., 2013) dataset contains images of 102 model variants with 100 images per class.

The **VGG Flowers (Flower)** (Nilsback and Zisserman, 2008) dataset is comprised of natural images of 102 flower categories. The number of images in each class ranges from 40 to 258.

The **CUB-200-2011 (CUB)** (Welinder et al., 2011) dataset is based on the fine-grained classification of 200 different bird species.

The **Describable Textures (DTD)** (Cimpoi et al., 2014) dataset comprises 5,640 images organised according to a list of 47 texture categories (classes) inspired by human perception.

The **Quick Draw (QDraw)** (Jongejan et al., 2016) is a dataset consisting of 50 million black-and-white drawings of 345 categories (classes), contributed by players of the game Quick, Draw!

The **Fungi** (Schroeder and Cui, 2018) dataset is comprised of around 100K images of 1,394 wild mushroom species, each forming a class.

The **MSCOCO (COCO)** (Lin et al., 2014) dataset is the train2017 split of the COCO dataset. COCO contains images from Flickr with 1.5 million object instances of 80 classes.

## A.2  MODEL ARCHITECTURES

Following (Lin et al., 2022), we employ optimised DNN architectures designed to be used in resource-limited IoT devices, including **MCUNet** (Lin et al., 2020), **MobileNetV2** (Sandler et al., 2018), and **ProxylessNASNet** (Cai et al., 2019). The DNN models are pre-trained using ImageNet (Deng et al., 2009). Specifically, the backbones of MCUNet (using the 5FPS ImageNet model), MobileNetV2 (with the 0.35 width multiplier), and ProxylessNAS (with a width multiplier of 0.3) have 23M, 17M, 19M MACs and 0.48M, 0.25M, 0.33M parameters, respectively. Note that MACs are calculated based on an input resolution of $128 \times 128$ with an input channel dimension of 3. The basic statistics of the three DNN architectures are summarised in Table 3.

## A.3  TRAINING DETAILS

We adopt a common training strategy to meta-train the pre-trained DNN backbones, which helps us avoid over-engineering the training process for each dataset and architecture (Hu et al., 2022).

Table 3: The statistics of our employed DNN architectures.

| Model | Param | MAC | # Layers | # Blocks |
|---|---|---|---|---|
| MCUNet | 0.46M | 22.5M | 42 | 13 |
| MobileNetV2 | 0.29M | 17.4M | 52 | 17 |
| ProxylessNASNet | 0.36M | 19.2M | 61 | 20 |

Specifically, we meta-train the backbone for 100 epochs. Each epoch has 2000 episodes/tasks. A warm-up and learning rate scheduling with cosine annealing are used. The learning rate increases from $10^{-6}$ to $5 \times 10^{-5}$ in 5 epochs. Then, it decreases to $10^{-6}$. We use SGD with momentum as an optimiser.

## A.4    DETAILS FOR EVALUATION SETUP

To evaluate the cross-domain few-shot classification performance, we sample 200 different tasks from the test split for each dataset. Then, as key performance metrics, we first use testing accuracy on unseen samples of a new domain as the target dataset. Note that the number of classes and support/query sets are sampled uniformly at random based on the dataset specifications. In addition, we analytically calculate the computation cost and memory footprint required for the forward pass and backward pass (*i.e.* model parameters, optimisers, activations). For the memory footprint of the backward pass, we include (1) model memory for the weights to be updated, (2) optimiser memory for gradients, and (3) activations memory for intermediate outputs for weights update. For the computational cost, as in (Xu et al., 2022), we report the number of MAC operations of the backward pass, which incurs $2\times$ more MAC operations than the forward pass (inference). Also, we measure latency and energy consumption to perform end-to-end training of a deployed DNN on the edge device. We deploy *TinyTrain* and the baselines on a tiny edge device, Pi Zero. To measure the end-to-end training time and energy consumption, we include the time and energy used to: (1) load a pre-trained model, (2) perform training using all the samples (*e.g.* 25) for a certain number of iterations (*e.g.* 40). For *TinyTrain*, we also include the time and energy to conduct a dynamic layer/ channel selection based on our proposed importance metric, by computing the Fisher information on top of those to load a model and fine-tune it. Regarding energy, we measure the total amount of energy consumed by a device during the end-to-end training process. This is performed by measuring the power consumption on Pi Zero using a YOTINO USB power meter and deriving the energy consumption following the equation: Energy = Power $\times$ Time.

**Further Details on Memory Usage.** There are several components that account for the memory footprint for the backward pass of training. Specifically, (F1) model weights and (F2) buffer space containing input and output tensors of a layer comprise the memory usage during the forward pass (*i.e.* inference). On top of that, during the backward pass (*i.e.* training), we also need to consider (B1) the model weights to be updated or accumulated gradients (*i.e.* a buffer space that contains newly updated weights or accumulated gradients from back-propagation), (B2) other optimiser parameters such as momentum values, (B3) values used to compute the derivatives of non-linear functions like ReLU from the last layer $L$ to a layer $i$ up to which we perform back-propagation, and (B4) inputs $x_i$ of the layers selected to be updated from the last layer $L$ to a layer $i$ up to which we back-propagate.

Regarding (B3), ReLU-type activation functions only need to store a binary mask indicating whether the value is smaller than zero or not. Hence, the memory cost of each non-linearity activation function based on ReLU is $|x_i|$ bits ($32\times$ smaller than storing the whole $x_i$), which is negligible. In our work, the employed network architectures (*e.g.* MCUNet, MobileNetV2, and ProxylessNASNet) rely on the ReLU non-linearity function. Regarding (B4), it is worth mentioning that when computing the gradient $g(W_i)$ given the inputs $(x_i)$ and the gradients $(g(x_{i+1}))$ to a $(i$-th) layer, we perform $g(W_i) = g(x_{i+1}).T * x_i$ to get gradient w.r.t the weights and $g(x_i) = g(x_{i+1}) * W_i$. Note that the intermediate inputs $(x_i)$ are only required to get the gradient of the weights $(g(W_i))$, meaning that the backward memory can be substantially reduced *if we do not update* the model weights $(W_i)$. This property is applicable to linear layers, convolutional layers, and normalisation layers as studied by Cai et al. (2020).

In our evaluation (§3.2), we conducted memory analysis to present the memory usage by taking into account both inference and backward-pass memory. We adopt the memory cost profiler used in prior work (Cai et al., 2020), which reuses the inference memory space during the backward pass wherever possible. Specifically, the memory space of (F2) can be overlapped with (B3) and (B4) as the buffer space for input and output tensors can be reused for intermediate variables of (B3) and (B4). On the other hand, the memory space for (B1) and (B2) cannot be overlapped with (F2) when the gradient accumulation is used as the system needs to retain the updated model weights and optimiser parameters throughout the training process. In addition, we would like to add that, depending on the hardware and deployment libraries, the model weights (F1) reside in the storage instead of being loaded on the main memory space. For example, on MCUs, model weights are stored on Flash (storage) and do not consume space on SRAM. Thus, we only include the model weights to be updated when calculating the memory usage for the backward pass.

## A.5 BASELINES

We include the following baselines in our experiments to evaluate the effectiveness of ***TinyTrain***.

**None.** This baseline does not perform any on-device training during deployment. Hence, it shows the accuracy drops of the DNNs when the model encounters a new task of a cross-domain dataset.

**FullTrain.** This method trains the entire backbone, serving as the strong baseline in terms of accuracy performance, as it utilises all the required resources without system constraints. However, this method intrinsically consumes the largest amount of system resources in terms of memory and computation among all baselines.

**LastLayer.** This refers to adapting only the head (*i.e.* the last layer or classifier), which requires relatively small memory footprint and computation. However, its accuracy typically is too low to be practical. Prior works (Ren et al., 2021; Lee and Nirjon, 2020) adopt this method to update the last layer only for on-device training.

**TinyTL** (Cai et al., 2020). This method proposes to add a small convolutional block, named the lite-residual module, to each convolutional block of the backbone network. During training, TinyTL updates the lite-residual modules while freezing the original backbone, requiring less memory and fewer computations than training the entire backbone. As shown in our results, TinyTrain requires the second largest amount of memory and compute resources among all baselines.

**SparseUpdate** (Lin et al., 2022). This method reduces the memory footprint and computation in performing on-device training. Memory reduction comes from updating selected layers in the network, followed by another selection of channels within the selected layers. However, SparseUpdate adopts a static channel and layer selection policy that relies on evolutionary search (ES). This ES-based selection scheme requires compute and memory resources that the extreme-edge devices can not afford. Even in the offline compute setting, it takes around 10 mins to complete the search.

## B DETAILS OF SAMPLING ALGORITHM DURING META-TESTING

### B.1 SAMPLING ALGORITHM DURING META-TESTING

We now describe the sampling algorithm during meta-testing that produces realistically imbalanced episodes of various ways and shots (*i.e.* K-way-N-shot), following Triantafillou et al. (2020). The sampling algorithm is designed to accommodate realistic deployment scenarios by supporting the various-way-various-shot setting. Given a data split (*e.g.* train, validation, or test split) of the dataset, the overall procedure of the sampling algorithm is as follows: (1) sample of a set of classes $\mathcal{C}$ and (2) sample support and query examples from $\mathcal{C}$.

**Sampling a set of classes.** First of all, we sample a certain number of classes from the given split of a dataset. The 'way' is sampled uniformly from the pre-defined range [5, MAX], where MAX indicates either the maximum number of classes or 50. Then, 'way' many classes are sampled uniformly at random from the given split of the dataset. For datasets with a known class organisation, such as ImageNet and Omniglot, the class sampling algorithm differs as described in (Triantafillou et al., 2020).

**Sampling support and query examples.** Having selected a set of classes, we sample support and query examples by following the principle that aims to simulate realistic scenarios with limited (*i.e.* few-shot) and imbalanced (*i.e.* realistic) support set sizes as well as to achieve a fair evaluation of our system via query set.

- **Support Set Size.** Based on the selected set of classes from the first step (*i.e.* sampling a set of classes), the support set is at most 100 (excluding the query set described below). The support set size is at least one so that every class has at least one image. The sum of support set sizes across all the classes is capped at 500 examples as we want to consider few-shot learning (FSL) in the problem formulation.

- **Shot of each class.** After having determined the support set size, we now obtain the 'shot' of each class.

- **Query Set Size.** We sample a class-balanced query set as we aim to perform well on all classes of samples. The number of minimum query sets is capped at 10 images per class.

### B.2 SAMPLE STATISTICS DURING META-TESTING

In this subsection, we present summary statistics regarding the support and query sets based on the sampling algorithm described above in our experiments. In our evaluation, we conducted 200 trials of experiments (200 sets of support and query samples) for each target dataset. Table 4 shows the average (Avg.) number of ways, samples, and shots of each dataset as well as their standard deviations (SD), demonstrating that the sampled target data are designed to be the challenging and realistic various-way-various-shot CDFSL problem. Also, as our system performs well on such challenging problems, we demonstrate the effectiveness of our system.

## C  FINE-TUNING PROCEDURE DURING META-TESTING

As we tackle realistic and challenging scenarios of the cross-domain few-shot learning (CDFSL) problem, the pre-trained DNNs can encounter a target dataset drawn from an unseen domain, where the pre-trained DNNs could fail to generalise due to a considerable shift in the data distribution.

Hence, to adjust to the target data distribution, we perform fine-tuning (on-device training) on the pre-trained DNNs by a few gradient steps while leveraging the data augmentation (as explained below). Specifically, the feature backbone as the DNNs is fine-tuned as our employed models are based on ProtoNet.

Our fine-tuning procedure during the meta-testing phase is similar to that of (Guo et al., 2020; Hu et al., 2022). First of all, as the support set is the only labelled data during meta-testing, prior work (Guo et al., 2020) fine-tunes the models using only the support set. For (Hu et al., 2022), it first uses data augmentation with the given support set to create a pseudo query set. After that, it uses the support set to generate prototypes and the pseudo query set to perform backpropagation using Eq. 1. Differently from (Guo et al., 2020),the fine-tuning procedure of (Hu et al., 2022) does not need to compute prototypes and gradients using the same support set using Eq. 1. However, Hu

Table 4: The summary statistics of the support and query sets sampled from nine cross-domain datasets.

|  | Traffic | Omniglot | Aircraft | Flower | CUB | DTD | QDraw | Fungi | COCO |
|---|---|---|---|---|---|---|---|---|---|
| Avg. Num of Ways | 22.5 | 19.3 | 9.96 | 9.5 | 15.6 | 6.2 | 27.3 | 27.2 | 21.8 |
| Avg. Num of Samples (Support Set) | 445.9 | 93.7 | 369.4 | 287.8 | 296.3 | 324.0 | 460.0 | 354.7 | 424.1 |
| Avg. Num of Samples (Query Set) | 224.8 | 193.4 | 99.6 | 95.0 | 156.4 | 61.8 | 273.0 | 105.5 | 217.8 |
| Avg. Num of Shots (Support Set) | 29.0 | 4.6 | 38.8 | 30.7 | 20.7 | 53.3 | 23.6 | 15.6 | 27.9 |
| Avg. Num of Shots (Query Set) | 10 | 10 | 10 | 10 | 10 | 10 | 10 | 10 | 10 |
| SD of Num of Ways | 11.8 | 10.8 | 3.4 | 3.1 | 6.6 | 0.8 | 13.2 | 14.4 | 11.5 |
| SD of Num of Samples (Support Set) | 90.6 | 81.2 | 135.9 | 159.3 | 152.4 | 148.7 | 94.8 | 158.7 | 104.9 |
| SD of Num of Samples (Query Set) | 117.7 | 108.1 | 34.4 | 30.7 | 65.9 | 8.2 | 132.4 | 51.8 | 114.8 |
| SD of Num of Shots (Support Set) | 21.9 | 2.4 | 14.9 | 14.9 | 10.5 | 24.5 | 17.0 | 8.9 | 20.7 |
| SD of Num of Shots (Query Set) | 0.0 | 0.0 | 0.0 | 0.0 | 0.0 | 0.0 | 0.0 | 0.0 | 0.0 |
| Num of Trials | 200 | 200 | 200 | 200 | 200 | 200 | 200 | 200 | 200 |

et al. (2022) simply fine-tune the entire DNNs without memory-and compute-efficient on-device training techniques, which becomes one of our baselines, FullTrain requiring prohibitively large memory footprint and computation costs to be done on-device during deployment. In our work, for all the on-device training methods including *TinyTrain*, we adopt the fine-tuning procedure introduced in (Hu et al., 2022). However, we extend the vanilla fine-tuning procedure with existing on-device training methods (*i.e.* LastLayer, TinyTL, SparseUpdate, which serve as the baselines of on-device training in our work) so as to improve the efficiency of on-device training on the extremely resource-constrained devices. Furthermore, our system, *TinyTrain*, not only extends the fine-tuning procedure with memory-and compute-efficient on-device training but also proposes to leverage data-efficient FSL pretraining to enable the first data-, memory-, and compute-efficient on-device training framework on edge devices.

## D    SYSTEM IMPLEMENTATION

The *offline* component of our system is built on top of PyTorch (version 1.10) and runs on a Linux server equipped with an Intel Xeon Gold 5218 CPU and NVIDIA Quadro RTX 8000 GPU. This component is used to obtain the pre-trained model weights, *i.e.* pre-training and meta-training. Then, the *online* component of our system is implemented and evaluated on Raspberry Pi Zero 2 and NVIDIA Jetson Nano, which constitute widely used and representative embedded platforms. Pi Zero 2 is equipped with a quad-core 64-bit ARM Cortex-A53 and limited 512 MB RAM. Jetson Nano has a quad-core ARM Cortex-A57 processor with 4 GB of RAM. Also, we do not use sophisticated memory optimisation methods or compiler directives between the inference layer and the hardware to decrease the peak memory footprint; such mechanisms are orthogonal to our algorithmic innovation and may provide further memory reduction on top of our task-adaptive sparse update.

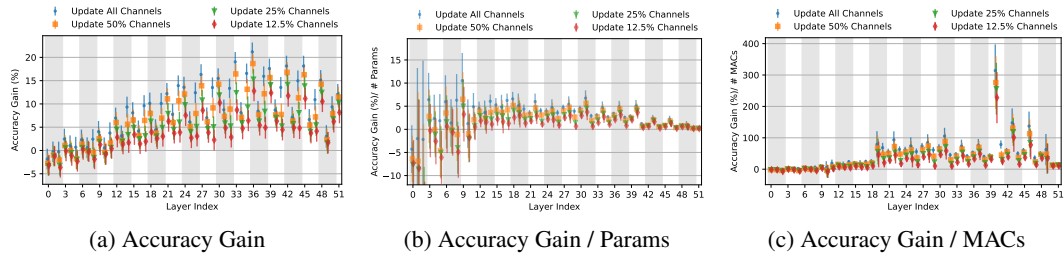

(a) Accuracy Gain       (b) Accuracy Gain / Params       (c) Accuracy Gain / MACs

Figure 7: Memory- and compute-aware analysis of **MobileNetV2** by updating four different channel ratios on each layer.

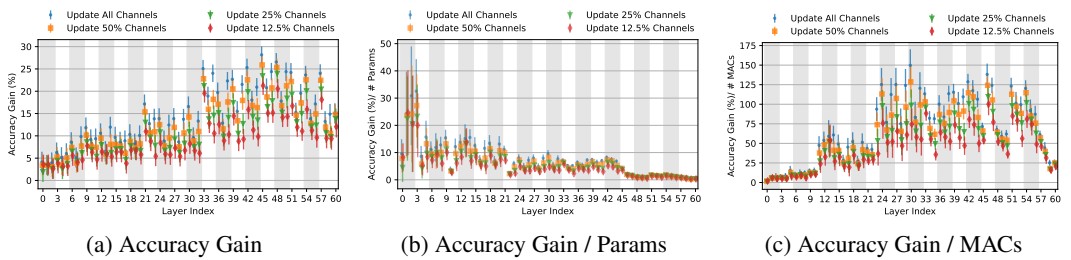

(a) Accuracy Gain       (b) Accuracy Gain / Params       (c) Accuracy Gain / MACs

Figure 8: Memory- and compute-aware analysis of **ProxylessNASNet** by updating four different channel ratios on each layer.

## E   ADDITIONAL RESULTS

In this section, we present additional results that are not included in the main content of the paper due to the page limit.

### E.1   MEMORY- AND COMPUTE-AWARE ANALYSIS

In §2.2, to investigate the trade-offs among accuracy gain, compute and memory cost, we analysed each layer's contribution (*i.e.* accuracy gain) on the target dataset by updating a single layer at a time, together with cost-normalised metrics, including accuracy gain *per parameter* and *per MAC operation* of each layer. MCUNet is used as a case study. Hence, here we provide the results of memory- and compute-aware analysis on the remaining architectures (MobileNetV2 and ProxylessNASNet) based on the Traffic Sign dataset as shown in Figure 7 and 8.

The observations on MobileNetV2 and ProxylessNASNet are similar to those of MCUNet. Specifically: (a) accuracy gain per layer is generally highest on the first layer of each block for both MobileNetV2 and ProxylessNASNet; (b) accuracy gain per parameter of each layer is higher on the second layer of each block for both MobileNetV3 and ProxylessNASNet, but it is not a clear pattern; and (c) accuracy gain per MACs of each layer has peaked on the second layer of each block for MobileNetV2, whereas it does not have clear patterns for ProxylessNASNet. These observations indicate a non-trivial trade-off between accuracy, memory, and computation for all the employed architectures in our work.

### E.2   PAIRWISE COMPARISON AMONG DIFFERENT CHANNEL SELECTION SCHEMES

Here, we present additional results regarding the pairwise comparison between our dynamic channel selection and static channel selections (*i.e.* Random and L2-Norm). Figure 9 and 10 show that the results of MobileNetV2 and ProxylessNASNet on the Traffic Sign dataset, respectively.

Similar to the results of MCUNet, the dynamic channel selection on MobileNetV2 and Proxyless-NASNet consistently outperforms static channel selections as the accuracy gain per layer differs by

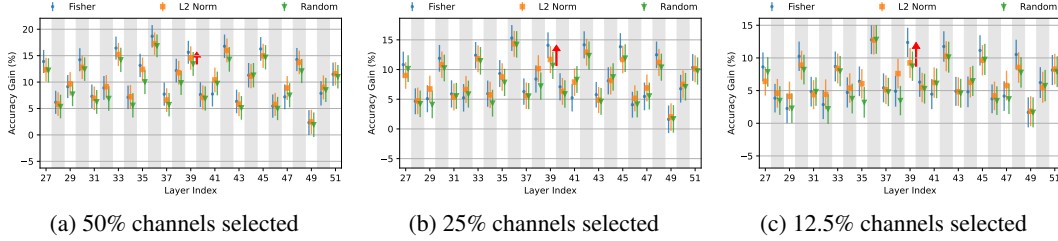

(a) 50% channels selected  (b) 25% channels selected  (c) 12.5% channels selected

Figure 9: The pairwise comparison between our dynamic channel selection and static channel selections (*i.e.* Random and L2-Norm) on **MobileNetV2**.

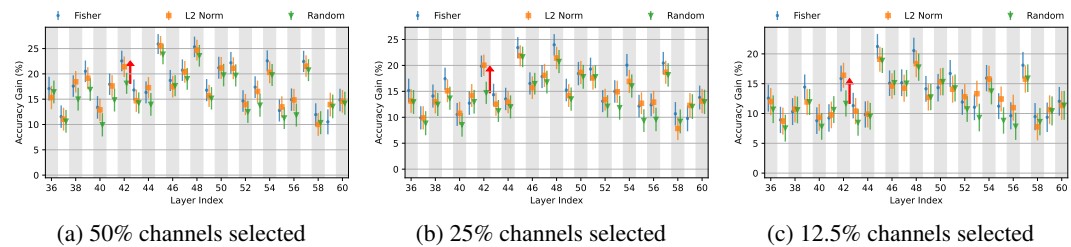

(a) 50% channels selected  (b) 25% channels selected  (c) 12.5% channels selected

Figure 10: The pairwise comparison between our dynamic channel selection and static channel selections (*i.e.* Random and L2-Norm) on **ProxylessNASNet**.

up to 5.1%. Also, the gap between dynamic and static channel selection increases as fewer channels are selected for updates.

### E.3   END-TO-END LATENCY BREAKDOWN OF *TinyTrain* AND *SparseUpdate*

In this subsection, we present the end-to-end latency breakdown to highlight the efficiency of our task-adaptive sparse update (*i.e.* the dynamic layer/channel selection process during deployment) by comparing our work (*TinyTrain*) with previous SOTA (*SparseUpdate*). We present the time to identify important layers/channels by calculating Fisher Potential (*i.e.* Fisher Calculation in Table 5 and 6) and the time to perform on-device training by loading a pre-trained model and performing backpropagation (*i.e.* Run Time in Tables 5 and 6).

In addition to the main results of on-device measurement on Pi Zero 2 presented in §3.2, we selected Jetson Nano as an additional device and performed experiments in order to ensure that our results regarding system efficiency are robust and generalisable across diverse and realistic devices. We used the same experimental setup (as detailed in §3.1 and §A.4) as the one used for Pi Zero 2.

As shown in Table 5 and 6, our experiments show that *TinyTrain* enables efficient on-device training, outperforming *SparseUpdate* by 1.3-1.7× on Jetson Nano and by 1.08-1.12× on Pi Zero 2 with respect to end-to-end latency. Moreover, Our dynamic layer/channel selection process takes around 18.7-35.0 seconds on our employed edge devices (*i.e.* Jetson Nano and Pi Zero 2), accounting for only 3.4-3.8% of the total training time of *TinyTrain*.

### E.4   IMPACT OF META-TRAINING

In this subsection, we present the complete results of the impact of meta-training. As discussed in §3.3, Figure 6a shows the average Top-1 accuracy with and without meta-training using MCUNet over nine cross-domain datasets. This analysis shows the impact of meta-training compared to conventional transfer learning, demonstrating the effectiveness of our FSL-based pre-training. However, it does not reveal the accuracy results of individual datasets and models. Hence, in this subsection, we present figures that compare Top-1 accuracy with and without meta-training for each architecture and dataset with all the on-device training methods to present the complete results of the impact of meta-training.

Table 5: The end-to-end latency breakdown of *TinyTrain* and SOTA on **Pi Zero 2**. The end-to-end latency includes time (1) to load a pre-trained model, (2) to perform training using given samples (*e.g.* 25) over 40 iterations, and (3) to calculate fisher information on activation (For *TinyTrain*).

| Model | Method | Fisher Calculation (s) | Run Time (s) | Total (s) | Ratio |
|---|---|---|---|---|---|
| MCUNet | SparseUpdate | 0.0 | 607 | 607 | 1.12× |
| | *TinyTrain* (Ours) | 18.7 | 526 | **544** | 1× |
| MobileNetV2 | SparseUpdate | 0.0 | 611 | 611 | 1.10× |
| | *TinyTrain* (Ours) | 20.1 | 536 | **556** | 1× |
| ProxylessNASNet | SparseUpdate | 0.0 | 645 | 645 | 1.08× |
| | *TinyTrain* (Ours) | 22.6 | 575 | **598** | 1× |

Table 6: The end-to-end latency breakdown of *TinyTrain* and SOTA on **Jetson Nano**. The end-to-end latency includes time (1) to load a pre-trained model, (2) to perform training using given samples (e.g., 25) over 40 iterations, and (3) to calculate fisher information on activation (For *TinyTrain*).

| Model | Method | Fisher Calculation (s) | Run Time (s) | Total (s) | Ratio |
|---|---|---|---|---|---|
| MCUNet | SparseUpdate | 0.0 | 1,189 | 1,189 | 1.3× |
| | *TinyTrain* (Ours) | 35.0 | 892 | **927** | 1× |
| MobileNetV2 | SparseUpdate | 0.0 | 1,282 | 1,282 | 1.5× |
| | *TinyTrain* (Ours) | 32.2 | 815 | **847** | 1× |
| ProxylessNASNet | SparseUpdate | 0.0 | 1,517 | 1,517 | 1.7× |
| | *TinyTrain* (Ours) | 26.8 | 869 | **896** | 1× |

Figures 11, 12, and 13 demonstrate the effect of meta-training based on MCUNet, MobileNetV2, and ProxylessNASNet, respectively, across all the on-device training methods and nine cross-domain datasets.

### E.5 ROBUSTNESS OF DYNAMIC CHANNEL SELECTION

As described in §3.3, to show how much improvement is derived from dynamically selecting important channels based on our method at deployment time, Figure 6b compares the accuracy of *TinyTrain* with and without dynamic channel selection, with the same set of layers to be updated within strict memory constraints using MCUNet. In this subsection, we present the full results regarding the robustness of our dynamic channel selection scheme using all the employed architectures and cross-domain datasets. Figures 14, 15, and 16 demonstrate the robustness of dynamic channel selection using MCUNet, MobileNetV2, and ProxylessNASNet, respectively, based on nine cross-domain datasets. Note that the reported results are averaged over 200 trials, and 95% confidence intervals are depicted.

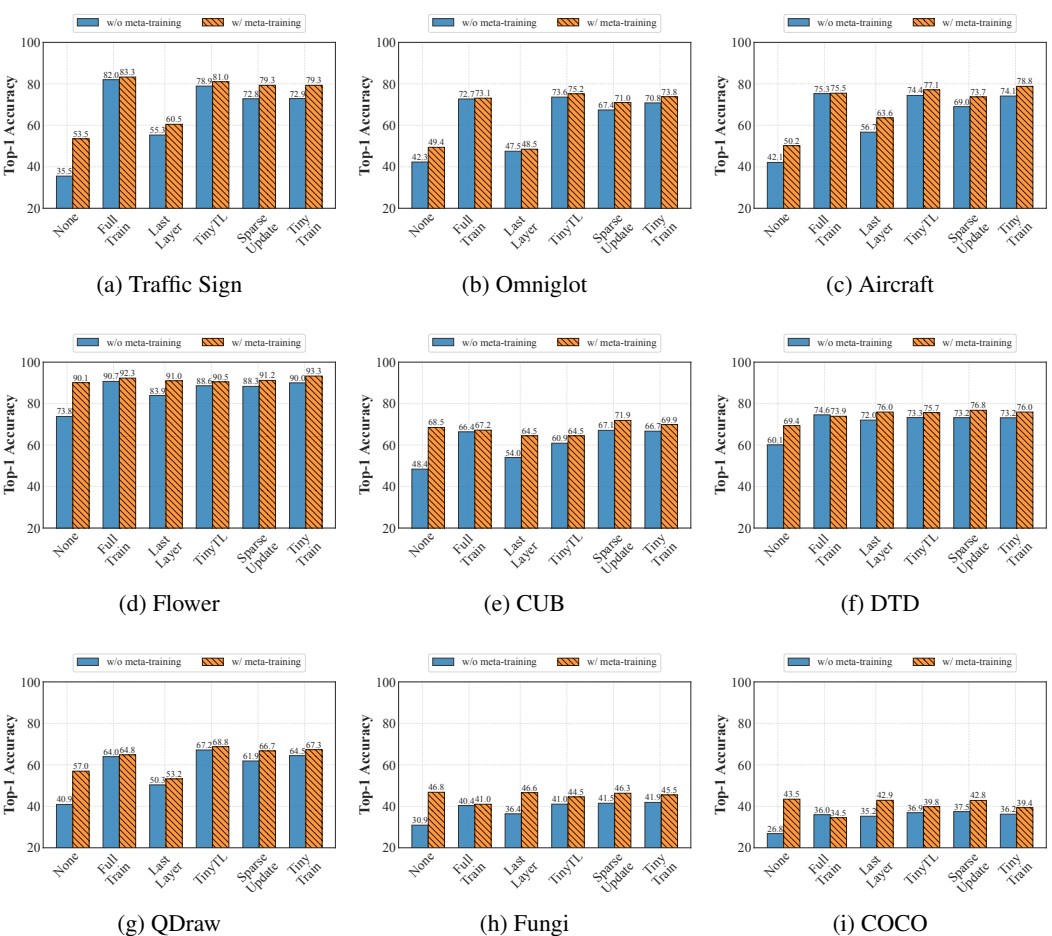

Figure 11: The effect of meta-training on **MCUNet** across all the on-device training methods and nine cross-domain datasets.

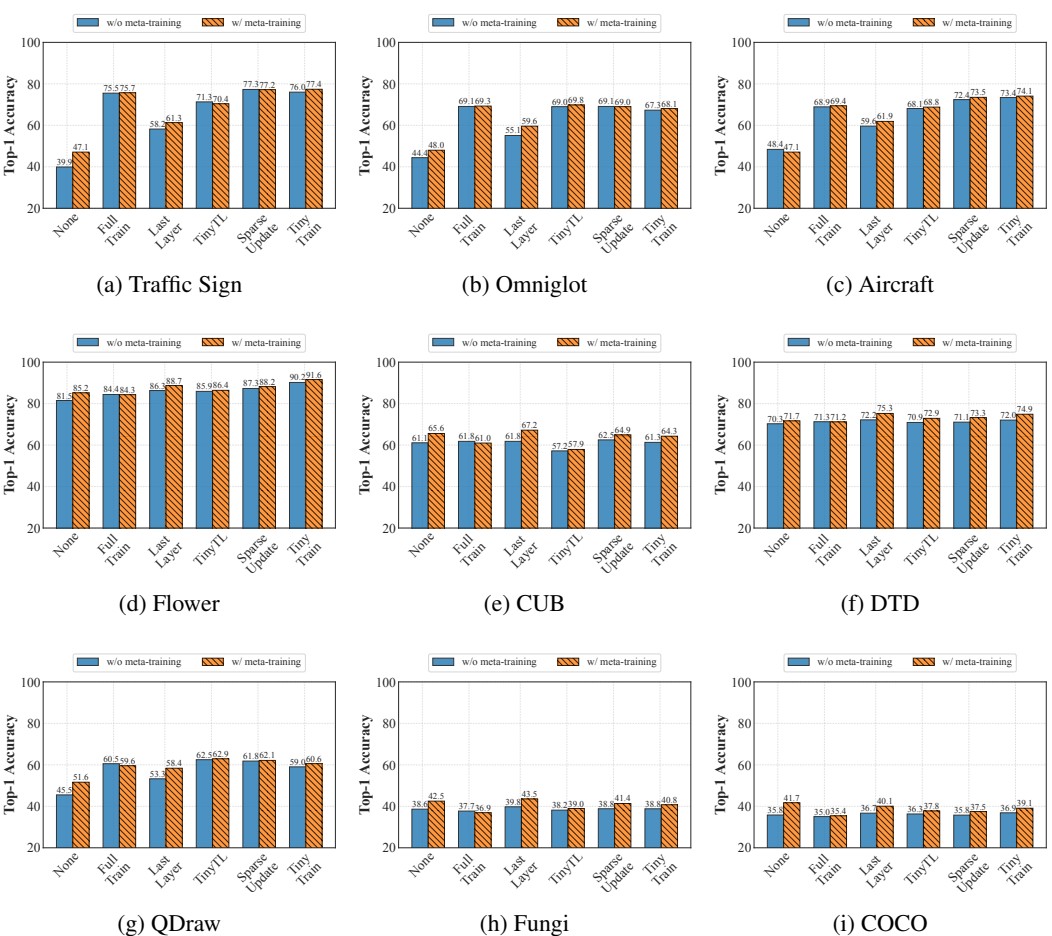

Figure 12: The effect of meta-training on **MobileNetV2** across all the on-device training methods and nine cross-domain datasets.

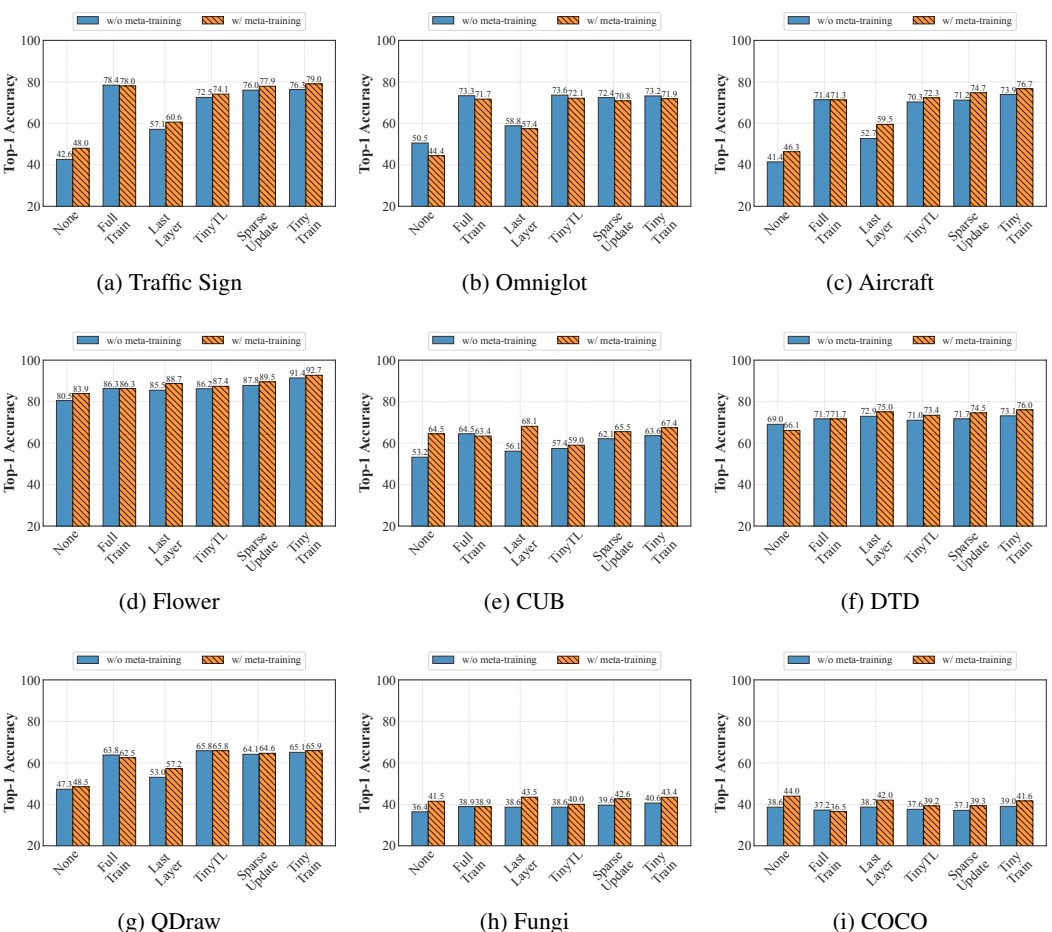

Figure 13: The effect of meta-training on **ProxylessNASNet** across all the on-device training methods and nine cross-domain datasets.

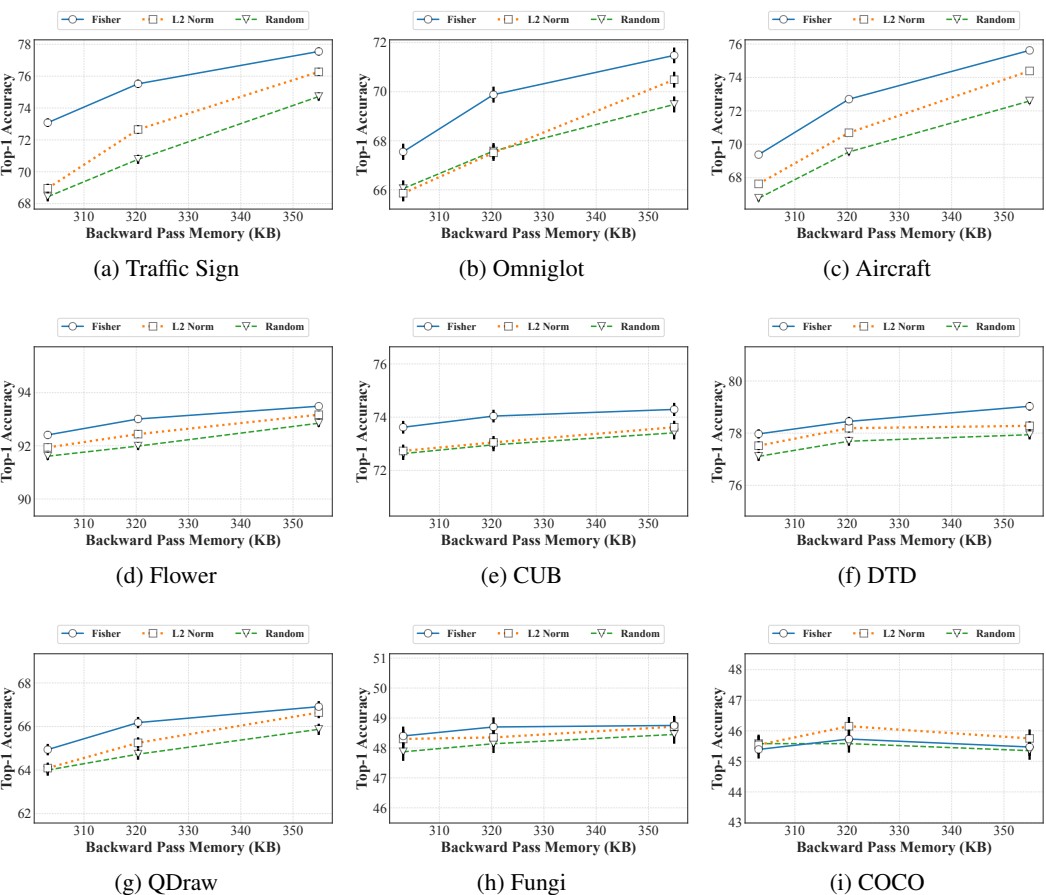

Figure 14: The effect of dynamic channel selection using **MCUNet** on nine cross-domain datasets.

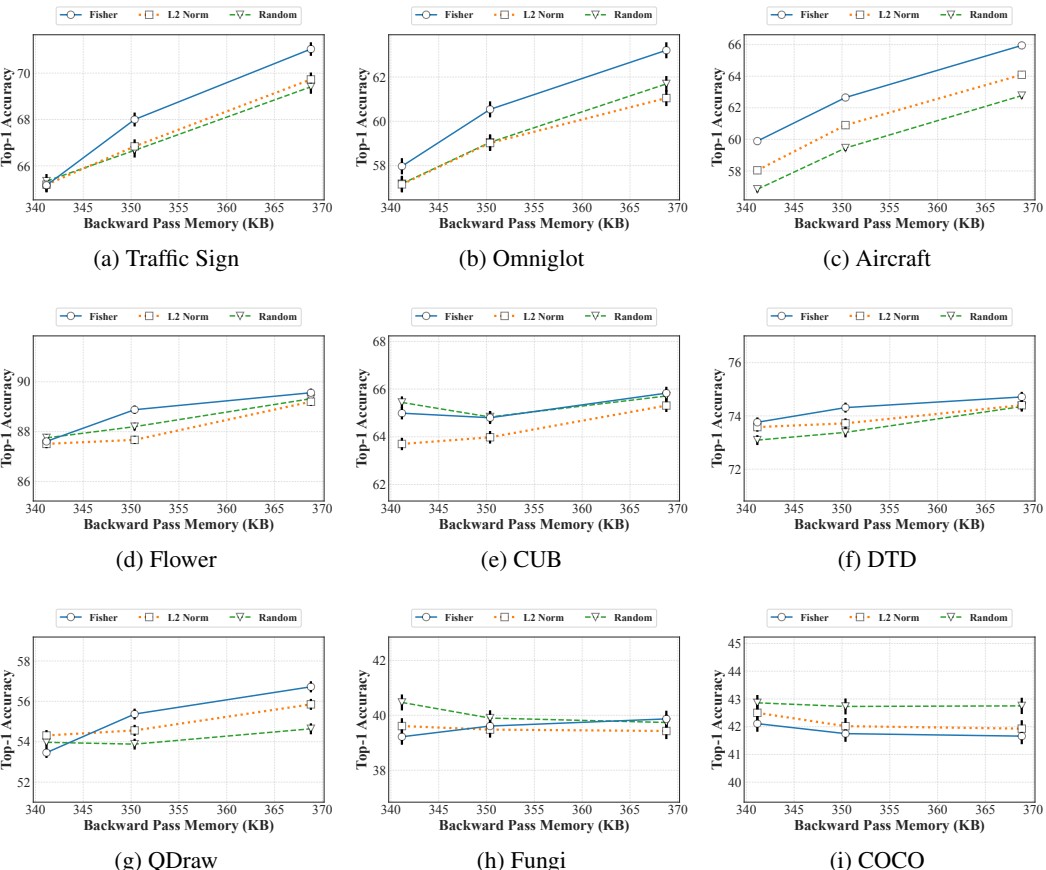

Figure 15: The effect of dynamic channel selection with **MobileNetV2** on nine cross-domain datasets.

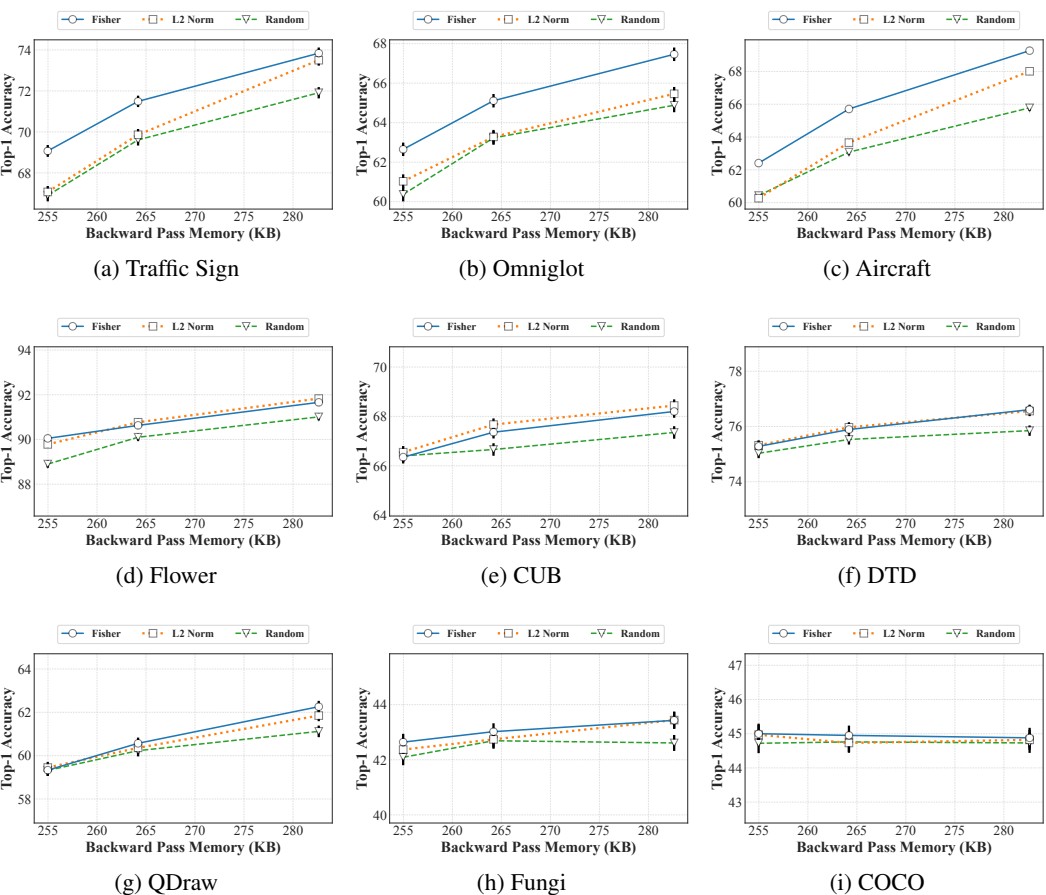

Figure 16: The effect of dynamic channel selection with **ProxylessNASNet** on all the datasets.

# F  FURTHER ANALYSIS AND DISCUSSION

## F.1  FURTHER ANALYSIS OF CALCULATING FISHER INFORMATION ON ACTIVATIONS

In this section, we describe how *TinyTrain* calculates the Fisher information on activations (the primary variable for our proposed multi-objective criterion) without incurring excessive memory overheads. Specifically, computing Fisher information on activations is designed to be within the memory and computation budget for a backward pass determined by hardware and users (e.g., in our evaluation, we use roughly 1 MB as a memory budget). Also, as described in Appendix A.4, the memory space used for saving intermediate variables can be overlapped with that of input/output tensors. As observed in prior works (Lin et al., 2022; 2021), the size of the activation is large for the first few layers and small for the remaining ones. Table 7 shows the saved activations' size to compute the backward pass up to the last $k$ blocks/layers. The sizes of saved activations are well within the peak memory footprint of input/output tensors (*i.e.* 640 KB for MCUNet, 896 KB for MobileNetV2, and 512 KB for ProxylessNASNet). Thus, the memory space of input/output tensors can be reused to store the intermediate variables required to calculate Fisher information on activations.

Also, we empirically demonstrated that important layers for a CDFSL task are located among those last few layers (as shown in Figure 3 for MCUNet, Figure 7 for MobileNetV2, and Figure 8 for ProxylessNASNet). A prior work (Lin et al., 2022) also observed the same trend. In our experiments, TinyTrain demonstrates that inspecting 30-44% of layers is enough to achieve SOTA accuracy, as shown in Table 1 in Section 3.2. Also, note that this process on edge devices is very swift as analysed in Section 3.3.

In addition, it is possible to further reduce the memory usage by optimising the execution scheduling during the forward pass (*e.g.* patch-based inference (Lin et al., 2021) or partial execution (Liberis and Lane, 2023)). This process trade-offs more computation for lower memory usage, consuming more time. However, this can reduce the peak memory to meet the constraints of the target platform. We leave this optimisation as future work.

Table 7: The total size of the saved activations in KB to compute the backward pass up to the last $k$ blocks/layers across three architectures used in our work.

| Last k Blocks | Last k Layers | MCUNet | MobileNetV2 | ProxylessNASNet |
|:---:|:---:|:---:|:---:|:---:|
| 6 | 18 | 479.0 | 432.9 | 299.3 |
| 5 | 15 | 392.3 | 325.7 | 241.5 |
| 4 | 12 | 281.0 | 218.4 | 171.6 |
| 3 | 9 | 191.6 | 148.5 | 118.0 |
| 2 | 6 | 135.9 | 101.6 | 89.1 |
| 1 | 3 | 80.3 | 54.7 | 60.3 |

## F.2  COST ANALYSIS OF META-TRAINING

In this subsection, we analyse the cost of meta-training, one of the major components of our FSL-based pre-training, in terms of the overall latency to perform meta-training. *TinyTrain*'s meta-training stage takes place offline (as illustrated in Figure 2) on a server equipped with sufficient computing power and memory (refer to §D for more details regarding hardware specifications used in our work) prior to deployment on-device. In our experiments, the offline meta-training on MiniImageNet takes around 5-6 hours across three architectures. However, note that this cost is small as meta-training needs to be performed *only once* per architecture. Furthermore, this cost is amortised by being able to reuse the same resulting meta-trained model across multiple downstream tasks (different target datasets) and devices, *e.g.* Raspberry Pi Zero 2 and Jetson Nano, while achieving significant accuracy improvements (refer to Table 1 and Figure 6a in the main manuscript and Figures 11, 12, and 13 in the appendix).

# G    EXTENDED RELATED WORK

## G.1    ON-DEVICE TRAINING

Scarce memory and compute resources are major bottlenecks in deploying DNNs on tiny edge devices. In this context, researchers have largely focused on optimising *the inference stage* (*i.e.* forward pass) by proposing lightweight DNN architectures (Gholami et al., 2018; Sandler et al., 2018; Ma et al., 2018), pruning (Han et al., 2016; Liu et al., 2020), and quantisation methods (Jacob et al., 2018; Krishnamoorthi, 2018; Rastegari et al., 2016), leveraging the inherent redundancy in weights and activations of DNNs. Also, researchers investigated on how to efficiently leverage heterogeneous processors (Jeong et al., 2022; Ling et al., 2021b;a), and offload computation (Yao et al., 2020). Driven by the increasing privacy concerns and the need for post-deployment adaptability to new tasks or users, the research community has recently turned its attention to enabling DNN *training* (*i.e.,* backpropagation having both forward and backward passes, and weights update) at the edge.

Researchers proposed memory-saving techniques to resolve the memory constraints of training (Sohoni et al., 2019; Chen et al., 2021; Pan et al., 2021; Evans and Aamodt, 2021; Liu et al., 2022). For example, gradient checkpointing (Chen et al., 2016; Jain et al., 2020; Kirisame et al., 2021) discards activations of some layers in the forward pass and recomputes those activations in the backward pass. Microbatching (Huang et al., 2019) splits a minibatch into smaller subsets that are processed iteratively, to reduce the peak memory needs. Swapping (Huang et al., 2020; Wang et al., 2018; Wolf et al., 2020) offloads activations or weights to an external memory/storage (*e.g.* from GPU to CPU or from an MCU to an SD card). Some works (Patil et al., 2022; Wang et al., 2022; Gim and Ko, 2022) proposed a hybrid approach by combining two or three memory-saving techniques. Although these methods reduce the memory footprint, they incur additional computation overhead on top of the already prohibitively expensive on-device training time at the edge. Instead, *TinyTrain* drastically minimises not only memory but also the amount of computation through its dynamic sparse update that identifies and trains only the most important layers/channels on-the-fly.

A few existing works (Lin et al., 2022; Cai et al., 2020; Profentzas et al., 2022; Qu et al., 2022) have also attempted to optimise both memory and computations, with prominent examples being *TinyTL* (Cai et al., 2020), *p-Meta* (Qu et al., 2022), and *SparseUpdate* (Lin et al., 2022). By selectively updating only a subset of layers and channels during on-device training, these methods effectively reduce both the memory and computation load. Nonetheless, as shown in §3.2, the performance of this approach drops dramatically (up to 7.7% for *SparseUpdate*) when applied at the extreme edge where data availability is low. This occurs because the approach requires access to the entire target dataset (*e.g. SparseUpdate* (Lin et al., 2022) uses the entire CIFAR-100 dataset (Krizhevsky et al., 2009)), which is unrealistic for such devices in the wild. More importantly, it requires a large number of epochs (*e.g. SparseUpdate* requires 50 epochs) to reach high accuracy, which results in an excessive training time of up to 10 days when deployed on extreme edge devices, such as STM32F746 MCUs. Also, these methods require running *a few thousands of* computationally heavy search (Lin et al., 2022) or pruning (Profentzas et al., 2022) processes on powerful GPUs to identify important layers/channels for each target dataset; as such, the current *static* layer/channel selection scheme cannot be adapted on-device to match the properties of the user data and hence remains fixed after deployment, leading to an accuracy drop. *p-Meta* enables pre-selected layer-wise updates learned during offline meta-training and dynamic channel-wise updates during online on-device training. However, as *p-Meta* requires additional learned parameters such as a meta-attention module identifying important channels for every layer, its computation and memory saving are relatively low. For example, *p-Meta* still incurs up to $4.7\times$ higher memory usage than updating the last layer only. In addition, *TinyTL* still demands excessive memory and computation (see §3.2). In contrast, *TinyTrain* enables data-, compute-, and memory-efficient training on tiny edge devices by adopting few-shot learning pre-training and dynamic layer/channel selection.

## G.2    FEW-SHOT LEARNING

Due to the scarcity of labelled user data on the device, developing Few-Shot Learning (FSL) techniques is a natural fit for on-device training (Hospedales et al., 2022). FSL methods aim to learn a target task given a few examples (*e.g.* 5-30 samples per class) by transferring the knowledge from large source data (*i.e.* meta-training) to scarcely annotated target data (*i.e.* meta-testing). Until now, several FSL schemes have been proposed, ranging from gradient-based (Finn et al., 2017; Antoniou

et al., 2018; Li et al., 2017), and metric-based (Snell et al., 2017; Sung et al., 2018; Satorras and Estrach, 2018) to Bayesian-based (Zhang et al., 2021). Recently, a growing body of work has been focusing on cross-domain (out-of-domain) FSL (CDFSL) (Guo et al., 2020). The CDFSL setting dictates that the source (meta-train) dataset drastically differs from the target (meta-test) dataset. As such, although CDFSL is more challenging than the standard in-domain (*i.e.* within-domain) FSL (Hu et al., 2022), it tackles more realistic scenarios, which are similar to the real-world deployment scenarios targeted by our work. In our work, we focus on realistic use-cases where the available source data (*e.g.* MiniImageNet (Vinyals et al., 2016)) are significantly different from target data (*e.g.* meta-dataset (Triantafillou et al., 2020)) with a few samples (5-30 samples per class), and hence incorporate CDFSL techniques into *TinyTrain*.

FSL-based methods only consider data efficiency and neglect the memory and computation bottlenecks of on-device training. Therefore, we explore joint optimisation of three major pillars of on-device training such as data, memory, and computation.

In addition, Un-/Self-Supervised Learning could be a potential solution to data scarcity issues. However, as investigated in (Liu et al., 2021), self-supervised learning in the presence of significant distribution shifts, as in the cross-domain tasks, could result in severe overfitting and insufficiency to capture the complex distribution of high-dimensional features in low-order statistics, leading to deteriorated accuracy. Further investigation could potentially reveal the feasibility of applying these techniques in cross-domain on-device training.

