# OpenReview forum: "TinyTrain: Deep Neural Network Training at the Extreme Edge"
_ICLR.cc/2024/Conference — Submitted to ICLR 2024_

### Official Review · Reviewer_k57W · 2023-10-29

**Soundness:** 3 good
**Presentation:** 2 fair
**Contribution:** 3 good
**Rating:** 8
**Confidence:** 4

**Summary:**

The authors present their approach to on-device training for edge devices, i.e., Jetson Nano and Raspberry Pi Zero. Their methodology includes meta-learning based offline pre-training and partial updates during online training with a channel selection strategy based on Fisher information and a multi-objective selection metric to jointly capture channel importance, memory footprint, and computational cost.

**Strengths:**

The authors contribute to the important area of on-device training at the edge. The novelties presented include the exploration of the capabilities of meta-learning as a means of offline pre-training, as well as a novel channel selection strategy that considers not only importance, but also goals such as computational complexity and memory footprint, which are generally important for deployment on resource-constrained targets. The authors provide solid empirical results by showing that their approach mostly outperforms other relevant state-of-the-art frameworks, namely TinyTL and MCUNet, on 9 different community datasets. Most interestingly, the authors also report results for latency and power consumption on two edge systems, the Jetson Nano and Pi Zero 2.

**Weaknesses:**

The title of the paper states that the authors want to explore training at the extreme edge, but in my opinion systems like the Pi Zero 2 and Jetson Nano are quite "large" compared to most Cortex-M based MCUs, which I would consider the extreme edge. Since such systems present unique challenges, e.g. they usually cannot run Linux but rely on RTOS like MBedOS or sometimes have limited floating-point support, I would have liked to see if the techniques proposed by the authors also work in such scenarios, similar to e.g. MCUNet. [1]


I found it hard to follow the paper at times, as it felt a bit unfocused, especially in Section 2, but the italicized summary provided at the end of each section definitely helped. Overall the paper is very dense (and many additional parts are left to the appendix), but it is not verbose.

[1] Lin, Ji, et al. "On-device training under 256kb memory." Advances in Neural Information Processing Systems 35 (2022): 22941-22954.

Some small points:
- Use ‘Sec.’ or ‘Section’ instead of ‘§’ (at least I am always stumbling at these points)
- Fig. 5a is missing.

**Questions:**

- Are the techniques presented by the authors applicable to Cortex-M based MCUs or only to the systems presented in this paper?
- Is meta-learning used only as part of pre-training or also for on-device training? It seems to me that classical transfer learning is done on-device. Do the authors think that on-device meta-learning techniques, e.g. few-shot learning, would be a feasible/reasonable approach?
- Does the layer selection process focus only on which channels to remove or also on how many channels to remove at any given time? If not, how do the authors determine how many channels of a tensor should be updated at any given time?
- How did you measure the energy consumption of your plattforms?
- I expected “FullTrain” to somehow act as an upper bound on predictive accuracy but it is not . can you elaborate on that in more detail?

**Details Of Ethics Concerns:**

The authors adequately address ethical and social implications of their work at the end of their manuscript.

---

> ### Author Response · Authors · 2023-11-22
>
> **Q1. Are the techniques presented by the authors applicable to Cortex-M based MCUs or only to the systems presented in this paper?**
>
> We have yet to deploy our method on ARM Cortex-M-based MCUs because no deep learning frameworks enable backpropagation to support dynamic channel selection. On-device training with the static channel selection is enabled based on the prior work [a], but its static configuration of layers and channels to be updated needs to be determined offline and cannot be dynamically changed online.
>
> However, we have implemented and successfully deployed our proposed method on Pi Zero 2 and Jetson Nano, demonstrating the feasibility of compute-, memory-, and data-efficient on-device training methodology on embedded systems. Similarly, our methodology can be applied to on-device training scenarios with extreme edge devices like Cortex-M-based MCUs given a backpropagation engine supporting dynamic layer/channel updates.
>
> [a] Ji Lin et al. "On-device training under 256kb memory." 36th Conference on Neural Information Processing Systems (NeurIPS 2022).
>
> **Q2. Is meta-learning used only as part of pre-training or also for on-device training? It seems to me that classical transfer learning is done on-device. Do the authors think that on-device meta-learning techniques, e.g. few-shot learning, would be a feasible/reasonable approach?**
>
> Yes, the meta-training is part of the few-shot learning (FSL) based pre-training as described in Section 2.1. Also, meta-training takes place offline before the online deployment stage (please refer to Section F.2 for the cost analysis of meta-training taking place offline).
>
> TinyTrain performs meta-learning, not transfer learning, on-device. as on-device adaptation is part of the process. Meta-learning (a.k.a. learning to learn) and/or few-shot learning consist of two stages, namely, meta-training and meta-testing. There are dedicated source and target datasets partitioned into sub-tasks employed by each of the two stages, as the idea is to learn in the inner loop one sub-task using the source dataset and evaluate the learned model on the target dataset. The meta-training phase aims to iteratively update the global weight initialisation through a range of sub-tasks, to be able to perform fast adaptation with a few samples when presented with new tasks. In other words, the learned DNNs are meta-tested, given a few examples of the meta-test (new) classes. In our case, this meta-test (adaptation + test) phase is performed on embedded systems. We will clarify this point further in our paper in Section 2.1.
>
> **Q3. Does the layer selection process focus only on which channels to remove or also on how many channels to remove at any given time? If not, how do the authors determine how many channels of a tensor should be updated at any given time?**
>
> In a nutshell, the dynamic layer/channel selection is about selecting which layers and channels to be trained on-device rather than selecting them to be removed. In detail, given (1) the model architecture (e.g., MCUNet’s backbone model contains 42 layers), (2) the memory budget, and (3) the compute budget, TinyTrain aims to select as many important channels and layers as possible without violating the memory and compute budgets.
>
> **Q4. How did you measure the energy consumption of your platforms?**
>
> Regarding energy consumption, we measure the total amount of energy consumed by a device during the end-to-end on-device training process. This is performed by measuring the power consumption on embedded devices using a YOTINO USB power meter and deriving the energy consumption following the equation: Energy = Power x Time. Then, to measure the end-to-end training time, we include the time elapsed to (1) load a pre-trained model and (2) perform training using all the samples (e.g. 25) for a certain number of iterations (e.g. 40). In particular, for reporting the time used for TinyTrain, we also include the time to conduct a dynamic layer/channel selection based on our proposed multi-criterion metric by computing the Fisher information on top of those to load a model and fine-tune it. Please refer to Section A.4 for more details.

---

> ### Author Response · Authors · 2023-11-22
>
> **Q5. I expected “FullTrain” to somehow act as an upper bound on predictive accuracy but it is not. can you elaborate on that in more detail?**
>
> In the case of a conventional transfer learning setup where the method is given at least a few thousand labelled samples, FullTrain would serve as the upper bound on predictive accuracy. However, we deal with the more challenging problem of cross-domain (out-of-domain) few-shot learning where the total number of labelled samples is at most 500. Hence, FullTrain, i.e., updating the whole network, may result in overfitting, leading to slightly lower accuracy. However, FullTrain still serves as a strong baseline. In addition, the improved accuracy of TinyTrain is largely derived from the effectiveness of the offline meta-training. As shown in Figure 6a, TinyTrain does not outperform FullTrain without meta-training. Then, with meta-training, TinyTrain starts to exceed FullTrain’s predictive accuracy. Updating the full networks could overfit due to memorization overfit or learner overfit as in [f]. It is possible the current setting is in favour of partial updates rather than FullTrain. We leave this as part of future investigation as it is more in the scope of meta-learning research.
>
> [f] Janarthanan Rajendran et al. "Meta-Learning Requires Meta-Augmentation" 34th Conference on Neural Information Processing Systems (NeurIPS 2020)
>
> **Minor Comments: Use ‘Sec’ or ‘Section’ instead of §, and Figure 5a is missing**
>
> Following the reviewer’s suggestion, we will use Sec. instead of § when referring to a section. Also, we will revise the captions of Figure 5.

---

### Official Review · Reviewer_T24f · 2023-10-29

**Soundness:** 2 fair
**Presentation:** 3 good
**Contribution:** 2 fair
**Rating:** 5
**Confidence:** 5

**Summary:**

The authors proposed a few-shot learning pipeline for edge devices, where both the data and the resources are limited during the training. The authors proposed a structured sparse-updating method that can dynamically select the critical layers/channels for each new learning task. During the few-shot learning, only these critical layers/channels will be updated. Therefore, both the training time and the energy cost are significantly reduced, which is also verified by their deployment results.

**Strengths:**

- The paper is well-motivated. Learning a new unseen task on edge devices with customized user data faces the shortage from data and computing resources, as the single user often has a limited labor resource to collect and label samples and the edge devices often have a limited on-device memory and computing power. How to achieve a comparable performance with limited training samples on resource-constrained edge device is important to the scenarios where the user has privacy concerns and the user data can only be processed locally.
- The authors conducted extensive experiments on different benchmarks. The authors also deployed their models on Pi Zero and Jetson Nano to measure the real training latency and energy cost, which is especially encouraged.

**Weaknesses:**

- The paper has a limited novelty. Although this paper proposed a different metric to select adaptation-critical weights, one of the related work "p-Meta: Towards On-device Deep Model Adaptation" had studied the same problem settings, i.e., how to train the model on new tasks given limited samples and limited computing resources. The proposed pipeline in p-Meta also consists of two stages, meta-training stages in the cloud and few-shot leaning stages on edge devices.

- I had some other concerns about how the authors computed the memory footprint in back-propagation. Since for a back-propagation, we must first conduct the forward pass. The peak memory consumption occurred during the backward pass at a certain layer only if it was larger than the peak memory required by the forward pass. Could you please elaborate how the memory number was calculated in Tab.2. At which layer during the back-propagation it reached the peak memory?
The reason why I had the concerns above was that 8-bit MCUNet had a 0.49MB peak memory in the forward pass, as reported in Fig.8 in https://arxiv.org/pdf/2007.10319.pdf. Then, the peak memory of training MCUNet in 32-bit floating point must be higher than 1.96MB (0.49*4), since you may also want to store some other intermediate values for backward. Unless the authors conducted low-precision training or used different architectures.

- I noticed that the authors used momentum SGD during on-device few-shot learning. What was the memory consumption from momentums? If these parameters took the majority, the authors should report a more fair comparison using vanilla SGD.

**Questions:**

See Weaknesses

---

> ### Author Response · Authors · 2023-11-23
>
> **Comment1. The paper has a limited novelty. Although this paper proposed a different metric to select adaptation-critical weights, one of the related work "p-Meta: Towards On-device Deep Model Adaptation" had studied the same problem settings, i.e., how to train the model on new tasks given limited samples and limited computing resources. The proposed pipeline in p-Meta also consists of two stages, meta-training stages in the cloud and few-shot learning stages on edge devices.**
>
> The objective of p-Meta and our work is the same, i.e., achieving efficient and accurate on-device training. However, those two approaches have considerable differences, providing unique benefits to TinyTrain over p-Meta. p-Meta uses a meta attention module that identifies the more important layers/channels, which shows the memory/computation reduction but does not take memory and computational costs as an optimisation goal explicitly. Conversely, we design a multi-criterion metric that explicitly takes the memory and computation costs into account when identifying which layers and channels to update, which would help our method select important layers/channels with less memory and computational costs, leading to a higher memory reduction.
>
> Secondly, p-Meta’s updated layers/channels change for each sample and each iteration, which could result in very frequent write operations on storage. Considering the real-world deployment scenario, especially for microcontrollers (MCUs) where the write operation on the storage (Flash) is two orders of magnitude more costly compared to the read operation [e], frequent changes of the update configuration would harm the system efficiency. On the other hand, TinyTrain performs only once the layer/channel selection at the beginning of the on-device training to select which layers/channels to update for a target task and they are fixed during on-device training, which suits deployment scenarios on MCUs well. We will clarify these points in Section 4 and Section G.
>
> [e] Filip Svoboda et al. 2022. Deep Learning on Microcontrollers: A Study on Deployment Costs and Challenges. In Proceedings of the 2nd European Workshop on Machine Learning and Systems (Rennes, France) (EuroMLSys ’22).
>
> **Comment2. Could you please elaborate how the memory number was calculated in Tab.2. At which layer during the back-propagation it reached the peak memory? The reason why I had the concerns above was that 8-bit MCUNet had a 0.49MB peak memory in the forward pass, as reported in Fig.8 in https://arxiv.org/pdf/2007.10319.pdf. Then, the peak memory of training MCUNet in 32-bit floating point must be higher than 1.96MB (0.49*4), since you may also want to store some other intermediate values for backward. Unless the authors conducted low-precision training or used different architectures.**
>
> The model mentioned in the comment above is the largest MCUNet architecture: it consumes 1.96 MB of memory in the forward pass alone. We use a different version of MCUNet, which is much smaller than the one above. Its peak memory use for the forward pass is 0.625 MB rather than 1.96 MB as the reviewer specified. Please refer to Section A.2 for more details regarding the model architectures used in our study.
> > Following (Lin et al., 2022), we employ optimised DNN architectures designed to be used in resource-limited IoT devices, including MCUNet (Lin et al., 2020), MobileNetV2 (Sandler et al., 2018), and ProxylessNASNet (Cai et al., 2019). The DNN models are pre-trained using ImageNet (Deng et al., 2009). Specifically, the backbones of MCUNet (using the 5FPS ImageNet model), MobileNetV2 (with the 0.35 width multiplier), and ProxylessNAS (with a width multiplier of 0.3) have 23M, 17M, 19M MACs and 0.48M, 0.25M, 0.33M parameters, respectively. Note that MACs are calculated based on an input resolution of 128 × 128 with an input channel dimension of 3. The basic statistics of the three DNN architectures are summarised in Table 3.
>
> We will clarify this point in Sections 3.1 and A.2.

---

> > ### Author Response · Authors · 2023-11-23
> >
> > **Comment3. I noticed that the authors used momentum SGD during on-device few-shot learning. What was the memory consumption from momentums? If these parameters took the majority, the authors should report a more fair comparison using vanilla SGD.**
> >
> > As the reviewer suggested, we provide a detailed breakdown of the memory footprint of different on-device training methods using two optimisers, ADAM and SGD (see Table A). In our evaluation, we employed ADAM optimiser during meta-testing as it achieves the highest accuracy compared to other optimiser types. Our memory breakdown shows that the majority of the memory footprint is due to the activation memory of the forward pass. In detail, activation memory reached its peak during the forward pass as the saved intermediate activations do not put additional memory overhead as the inference memory space can be reused during the backward pass to save the intermediate activations. Then, the optimiser incurs memory overhead. As the reviewer pointed out, the optimiser type affects the total memory footprint and associated memory reduction ratio. Yet, TinyTrain still outperforms all the on-device training baselines. We will incorporate this analysis in the additional result section in Section E.
> >
> > Table A. The detailed breakdown of the memory footprint of on-device training methods based on MCUNet according to different optimisers.
> > |  |  | **ADAM** |  | **SGD** |  |
> > |---|---|---|---|---|---|
> > | **Method** | **Memory Type** | **Memory** | **Ratio** | **Memory** | **Ratio** |
> > | LastLayer | Updated Weights | 0.35 MB | - | 0.35 MB | - |
> > |  | Optimiser | 1.05 MB | - | 0.35 MB | - |
> > |  | Activation | 0.63 MB | - | 0.63 MB | - |
> > |  | Total | 2.03 MB | 2.27x | 1.33 MB | 1.75x |
> > | SparseUpdate | Updated Weights | 0.20 MB | - | 0.20 MB | - |
> > |  | Optimiser | 0.60 MB | - | 0.20 MB | - |
> > |  | Activation | 0.63 MB | - | 0.63 MB | - |
> > |  | Total | 1.43 MB | 1.59x | 1.03 MB | 1.35x |
> > | TinyTrain | Updated Weights | 0.07 MB | - | 0.07 MB | - |
> > |  | Optimiser | 0.20 MB | - | 0.07 MB | - |
> > |  | Activation | 0.63 MB | - | 0.63 MB | - |
> > |  | Total | 0.89 MB | 1.00x | 0.76 MB | 1.00x |

---

### Official Review · Reviewer_aEBW · 2023-10-30

**Soundness:** 2 fair
**Presentation:** 3 good
**Contribution:** 2 fair
**Rating:** 5
**Confidence:** 4

**Summary:**

This paper develops TinyTrain as an on-device training approach that addresses the challenges of data scarcity and resource constraints in the context of IoT devices and microcontroller units (MCUs). Traditional methods neglect the data scarcity issue, require long training times, or result in significant accuracy loss. However, TinyTrain introduces a task-adaptive sparse-update method that dynamically selects layers and channels based on a multi-objective criterion. This approach considers user data, memory, and compute capabilities, leading to improved accuracy on unseen tasks with reduced computation and memory requirements. The proposed TinyTrain fits the memory constraints of MCU-grade platforms and ensures practical feasibility.

**Strengths:**

The proposed TinyTrain presents a practical solution for training models on resource-constrained edge devices, helping improve the efficiency of on-device model training.

**Weaknesses:**

(1) The experiments mainly concentrate on image classification with a limited number of training epochs, so it would be valuable to also evaluate the performance of the proposed methods on segmentation and detection tasks. These tasks often involve more complex data and require more extensive training. Examining the effectiveness of the proposed methods in such scenarios would provide a more comprehensive assessment of their applicability and overall performance.

(2) To gain a better understanding of the practical implications and efficiency of the proposed methods, it is suggested to include information on training/inference speed or time cost when applying the methods to downstream tasks. This would offer valuable insights into the computational efficiency and scalability of the approach for real-time or time-sensitive applications.

**Questions:**

Please see the weakness part.

---

> ### Author Response · Authors · 2023-11-22
>
> **Comment1. The experiments mainly concentrate on image classification with a limited number of training epochs, so it would be valuable to also evaluate the performance of the proposed methods on segmentation and detection tasks. These tasks often involve more complex data and require more extensive training. Examining the effectiveness of the proposed methods in such scenarios would provide a more comprehensive assessment of their applicability and overall performance.**
>
> It is very challenging to support **even the forward pass** of such applications as segmentation and object detection due to the large input resolution. For instance, a single image with a resolution of 640x640x3 (a widely used resolution for the MSCOCO dataset [c,d]) accounts for 1.17 MB to store, exceeding the available memory on a microcontroller (1 MB if it is a high-end microcontroller), let alone the cost of DNN execution regarding the forward and backward passes. Investigating the effectiveness of TinyTrain on segmentation and detection can be interesting, however, the challenge regarding high-resolution image inputs needs to be dealt with first. Thus, we leave this task as future work. Also, we would like to highlight that within the current scope of the paper, we have evaluated the proposed method on 9 different vision datasets to showcase the superior performance TinyTrain achieves in image classification.
>
> In addition, we would like to clarify that the unique contribution of TinyTrain is the task-adaptive sparse update policy based on our proposed multi-objective criterion and dynamic layer/channel selection that guides the layer/channel selection process during deployment. We focus on a training framework that jointly addresses accuracy, memory, compute efficiency, as well as the scarcity of labelled data on user devices, which makes our work different from existing approaches in the sense that we consider realistic factors in real-world deployment comprehensively. Specifically, our online on-device learning stage leverages the offline meta-training which learns a sufficiently general representation in order to learn cross-domain tasks efficiently when given a few samples on-device (i.e. addressing (i) accuracy and (ii) data scarcity issues). Also, our proposed task-adaptive sparse update not only enables TinyTrain to automate the layer/channel selection process but also drastically reduce the memory and compute requirements of on-device training (i.e. addressing (iii) memory and (iv) computational efficiency issues) while achieving new SOTA accuracy results on nine cross-domain datasets across three architectures.
>
> [c] Tsung-Yi Lin et al. Microsoft COCO: Common Objects in Context. In European Conference on Computer Vision (ECCV), 2014.
>
> [d] Emanuel Ben-Baruch et al. Asymmetric Loss For Multi-Label Classification. Proceedings of the IEEE/CVF International Conference on Computer Vision (ICCV), 2021, pp. 82-9.

---

> > ### Author Response · Authors · 2023-11-22
> >
> > **Comment2. To gain a better understanding of the practical implications and efficiency of the proposed methods, it is suggested to include information on training/inference speed or time cost when applying the methods to downstream tasks. This would offer valuable insights into the computational efficiency and scalability of the approach for real-time or time-sensitive applications.**
> >
> > We have investigated the run-time efficiency of TinyTrain when it is applied to various downstream tasks (i.e. 9 cross-domain few-shot learning datasets) in Section 3.2. We have successfully deployed TinyTrain on two embedded devices such as Pi Zero 2 and Jetson Nano. In addition, we have measured the end-to-end on-device training time and energy consumption of the workloads of these downstream tasks and reported the results in Figure 5 and Section 3.2. The summary of results is written in the paper as follows.
> > > TinyTrain yields 1.08-1.12x and 1.3-1.7x faster on-device training than SOTA on Pi Zero 2 and Jetson Nano, respectively. Also, TinyTrain completes an end-to-end on-device training process within 10 minutes, an order of magnitude speedup over the two-hour training of conventional transfer learning, a.k.a. FullTrain on Pi Zero 2.
> >
> > Also, regarding the efficiency of our dynamic layer/channel selection process, we have examined the time it takes to identify important layers/channels and reported them (20-35 seconds on Pi Zero 2 and Jetson Nano, accounting for only 3.4-3.8% of the total training time of TinyTrain) in Sections 3.2 and E.3.
> >
> > Lastly, we have analysed the cost of meta-training that needs to be done offline before deploying the model on resource-constrained devices in Section F.2. We wrote as follows.
> > > In our experiments, the offline meta-training on MiniImageNet takes around 5-6 hours across three architectures. However, note that this cost is small as meta-training needs to be performed only once per architecture. Furthermore, this cost is amortised by being able to reuse the same resulting meta-trained model across multiple downstream tasks (different target datasets) and devices, e.g. Raspberry Pi Zero 2 and Jetson Nano, while achieving significant accuracy improvements (refer to Table 1 and Figure 6a in the main manuscript and Figures 11, 12, and 13 in the appendix).

---

### Official Review · Reviewer_pF1F · 2023-10-31

**Soundness:** 2 fair
**Presentation:** 2 fair
**Contribution:** 2 fair
**Rating:** 3
**Confidence:** 4

**Summary:**

TinyTrain offers an approach to on-device training tailored to IoT and MCU devices, which are typically faced with limited memory and computational resources. By adopting a task-adaptive sparse-update method, this system aims to train neural network models more efficiently, addressing data scarcity and aiming to curtail training durations. TinyTrain claims to achieve a 3.6-5.0% improvement in accuracy compared to conventional methods and boasts of a 9.5× faster training speed.

**Strengths:**

1. This paper proposes a task-specific sparse update for the model. This dynamic sparse-update configuration is better in model accuracy than that method that adopts static configuration.

2. This paper enhances on-device learning by incorporating a few-shot learning scheme to let it be sample-efficient.

**Weaknesses:**

1. The text in the figures is too small and hard to read.

2. Some figures are not well-presented.

3. The design seems to have little to do with the observations of the paper.

**Questions:**

1. The text in almost all the figures is too small and pretty hard to read. The authors should make sure the figures are easy to read.

2. In Figure 3, the authors want to claim that i) Accuracy gain per layer is generally highest on the first layer of each block, ii) Accuracy gain per parameter and computation cost of each layer is higher on the second layer of each block. While I appreciate these interesting observations, I do have some concerns and comments:
(1) The authors have some observations related to the position of a layer in the block. However, all we can see about the layer position in the figures is the layer index in the whole model. I have no idea which layers are the first/second layers in the block.
(2) The authors say that the second observation is not a clear pattern. If it is not a clear pattern, why bother writing down this uncertain observation without any further explanation?
(3) What are the accuracy gain per parameter and accuracy gain per MAC? Are these widely used terms? May you briefly introduce them before using them?

3. This paper gives out some observations. However, they seem to have limited contribution to the design. The design uses a gradient-based metric to indicate the importance of a layer/channel, which does not consider the layer position in the block.

4. There is already a paper [1] that jointly considers the layer importance and computation cost when selectively updating some layers of a model, which is similar to the idea of this paper. I have concerns about the novelty of this paper.


[1] Yue Wang, Ziyu Jiang, Xiaohan Chen, Pengfei Xu, Yang Zhao, Yingyan Lin, and Zhangyang Wang. 2019. E2-train: training state-of-the-art CNNs with over 80% energy savings. Proceedings of the 33rd International Conference on Neural Information Processing Systems. Curran Associates Inc., Red Hook, NY, USA, Article 462, 5138–5150.

---

> ### Author Response · Authors · 2023-11-23
>
> **Comment1. The text in the figures is too small and hard to read.**
>
> We will re-draw the figures by increasing the font size.
>
> **Comment2. The authors have some observations related to the position of a layer in the block. However, all we can see about the layer position in the figures is the layer index in the whole model. I have no idea which layers are the first/second layers in the block.**
>
> We will re-draw the figures so that readers can see each block in the networks. Specifically, blocks in the figures are coloured in either white or grey so that each block is visually separated. Also, to help readers easily identify which layer is the first and which is the second in the block, we put the layer index in one block as an example.
>
> **Comment3. The authors say that the second observation is not a clear pattern. If it is not a clear pattern, why bother writing down this uncertain observation without any further explanation?**
>
> In this work, we make the following observations as quoted below.
> > (1) the peak point of accuracy gain occurs at the first layer of each block (pointwise convolutional layer) (Figure 3a), (2) the accuracy gain per parameter and computation cost occurs at the second layer of each block (depthwise convolutional layer) (Figures 3b and 3c).
>
> A key takeaway message of the second observation is to indicate that selecting the first layer of each block (which would be the case based on the contribution analysis in the prior work [a] and our first observation) may not always be an optimal choice for a layer/channel selection of on-device training when taking into account both accuracy and system aspects such as memory and computation. In other words, it indicates a non-trivial trade-off between accuracy, memory, and computation, which motivates us to propose our multi-objective criterion for an effective layer/channel selection that jointly considers all three aspects above. We will clarify this point in Section 2.2.
>
> [a] Ji Lin et al. "On-device training under 256kb memory." 36th Conference on Neural Information Processing Systems (NeurIPS 2022).
>
> **Comment4. What are the accuracy gain per parameter and accuracy gain per MAC? Are these widely used terms? May you briefly introduce them before using them?**
>
> We introduced two new terms, (1) accuracy gain per parameters and (2) accuracy gain per MACs, in order to take into account the system aspects to enable resource-aware layer/channel selection for on-device training. First, accuracy gain per parameter indicates how much accuracy gain we can obtain by updating a single layer at a time divided by the number of parameters in the layer. Then, the accuracy gain per MAC represents how much accuracy gain we can obtain by updating a single layer at a time divided by the number of MACs in the layer. We will introduce and explain those terms in detail before we present our observations in Section 2.2.
>
> **Comment5. This paper gives out some observations. However, they seem to have limited contribution to the design. The design uses a gradient-based metric to indicate the importance of a layer/channel, which does not consider the layer position in the block.**
>
> As explained in our response to Comment 3 above, the observations made in this work directly inspired us to propose the multi-objective criterion, which is the first component of our task-adaptive sparse-update method. As the multi-objective criterion considers the importance of each layer and channel as well as its memory and computation overheads, it allows us to encompass both accuracy and efficiency aspects.
>
> Having established the multi-objective criterion, we design the dynamic layer/channel selection scheme, the other component of our task-adaptive sparse update. Our observations may not directly affect the design of the dynamic layer/channel selection scheme; however, it leverages the multi-objective criterion to quickly identify the layers/channels that are important as well as memory- and compute-efficient on the fly during deployment given a target task, enabling our on-device training framework to be task adaptive. Also, in our dynamic layer/channel selection scheme, although we do not explicitly design our method to select each block's first or second layer, we observed that the first/second layers in a block are typically selected in our evaluation.

---

> > ### Author Response · Authors · 2023-11-23
> >
> > **Comment6. There is already a paper [1] that jointly considers the layer importance and computation cost when selectively updating some layers of a model, which is similar to the idea of this paper. I have concerns about the novelty of this paper.**
> >
> > We will clarify our contribution over E2-train [b] in Sections 4 and G. However, here are the salient points highlighting the superiority of TinyTrain over E2-train: E2-train introduces a trainable gating function. This consists of an RNN model per layer block - although the Appendix says that all RNNs share weights to save memory footprint. The RNNs are trained alongside the main on-device training process. The utilisation of RNN-based trainable gating functions i) adds overhead to the training stage in terms of training time and ii) opens up questions with respect to scalability (The paper does not report training time results, only energy).
> >
> > Also, if E2-train's selective layer update (SLU) was to be trained during the offline meta-training stage of TinyTrain, it would require samples that are representative of the ones encountered upon deployment. In our FSL scenario, this cannot be the case by definition.
> >
> > E2-train performs selective updates only at the layer level. TinyTrain jointly considers both selective layer and channel updating, adding another dimension of adaptivity during training.
> >
> > [b] Yue Wang, Ziyu Jiang, Xiaohan Chen, Pengfei Xu, Yang Zhao, Yingyan Lin, and Zhangyang Wang. 2019. E2-train: training state-of-the-art CNNs with over 80% energy savings. Proceedings of the 33rd International Conference on Neural Information Processing Systems. Curran Associates Inc., Red Hook, NY, USA, Article 462, 5138–5150.

---

### Author Response · Authors · 2023-11-23
**A common response to all the reviewers**

Dear reviewers,

We sincerely appreciate all reviewers’ time and efforts in reviewing our paper and for the constructive feedback.
We appreciate all of you for your positive reviews and for **recognising** the strengths of our work:

- **pF1F:** Proposed sample-efficient method of on-device learning that improves accuracy over other methods.
- **aEBW:** Practical solution for training models on resource-constrained devices, improving the efficiency of on-device training.
- **T24f:** Well-motivated, extensive experiments on different benchmarks by deploying their models on two embedded devices.
- **k57W:** Novel contribution to the area of on-device training at the edge with solid empirical results.

We have addressed all the questions and comments raised by reviewers by providing **more clarifications**, presenting **new results**, generating a **new table**, and revising **tables and figures** during this rebuttal period. We summarise how we addressed the **reviewers’ main questions** as follows:

- **pF1F:** We clarified the observations made in our work and further elaborated on our design choices derived from the observations.
- **pF1F, T24f:** We clarified the unique contributions of our work compared to prior works, E2-train and p-Meta.
- **aEBW:** We described the unique challenge of different vision applications like segmentation and detection.
- **aEBW:** We elaborated on the end-to-end on-device training latency results on embedded devices and cost analysis of meta-training.
- **T24f:** We further explained the memory usage of the employed architectures in our work.
- **T24f:** We presented new results regarding the detailed memory footprint breakdown.
- **k57W:** We addressed the reviewer’s questions regarding the applicability of our work to different IoT platforms, meta-learning and transfer learning, and layer/channel selection.
- **k57W:** We further explained the energy measurement procedure and FullTrain’s accuracy compared to our work.

---

### Meta-Review · Area_Chair_KtaN · 2023-12-11

**Metareview:**

In this paper, the authors present a method for on-device training of neural networks in extremely compute and memory limited systems such as microcontroller (MCUs) and IoT devices. In particular, the authors introduce a sparse update learning method that trains convolutional neural networks with less memory and computation demands. The paper demonstrates the efficacy of their approach across multiple CNN architectures (MCUNet, MobileNet, and NASNet) across an array of datasets with an impressive reduction in memory and compute demands (e.g ~10x faster training speed).

The reviewers commented positively on how practical this method is for real world scenarios in which the memory and compute are extremely limited. The reviewers raised concerns on the limited focus of this method on image classification, the novelty with respect to prior work (p-Meta), the quality of the presentation, and the details about the methodology. The authors provided some responses indicating that they will improve the presentation of the paper in subsequent revisions and argued the novelty of their work with respect to prior papers.

The reviewers though remained divided about acceptance. Given the lack of consensus, I took the opportunity to read the paper more closely. I agreed with several reviewers that the quality of presentation needs serious revisions. Additionally, I would like to see more comparisons and detailed discussions of the comparisons with prior work. That said, my biggest concern is the limited applicability of this work to the ICLR conference. This work is very heavily application oriented and would better suit a computer vision community or a community interested in “on-edge” devices. I do not see as much general interest within the ICLR community for this area of work. For these reasons, this paper will not be accepted at this conference. I would encourage the authors to heavily revise this work based on the reviewer comments and resubmit this to a more specialized conference geared towards on-edge ML and computer vision.

**Justification For Why Not Higher Score:**

Limited interest to ICLR. Concerns about clarity of presentation.

**Justification For Why Not Lower Score:**

N/A

---

### Decision · Program_Chairs · 2024-01-16

Reject